# Understanding Sparse JL for Feature Hashing

**Meena Jagadeesan**[*]
Harvard University
Cambridge, MA 02138
mjagadeesan@college.harvard.edu

## Abstract

Feature hashing and other random projection schemes are commonly used to reduce the dimensionality of feature vectors. The goal is to efficiently project a high-dimensional feature vector living in $\mathbb{R}^n$ into a much lower-dimensional space $\mathbb{R}^m$, while approximately preserving Euclidean norm. These schemes can be constructed using sparse random projections, for example using a sparse Johnson-Lindenstrauss (JL) transform. A line of work introduced by Weinberger et. al (ICML '09) analyzes the accuracy of sparse JL with sparsity 1 on feature vectors with small $\ell_\infty$-to-$\ell_2$ norm ratio. Recently, Freksen, Kamma, and Larsen (NeurIPS '18) closed this line of work by proving a tight tradeoff between $\ell_\infty$-to-$\ell_2$ norm ratio and accuracy for sparse JL with sparsity 1.

In this paper, we demonstrate the benefits of using sparsity $s$ greater than 1 in sparse JL on feature vectors. Our main result is a tight tradeoff between $\ell_\infty$-to-$\ell_2$ norm ratio and accuracy for a general sparsity $s$, that significantly generalizes the result of Freksen et. al. Our result theoretically demonstrates that sparse JL with $s > 1$ can have significantly better norm-preservation properties on feature vectors than sparse JL with $s = 1$; we also empirically demonstrate this finding.

## 1 Introduction

Feature hashing and other random projection schemes are influential in helping manage large data [11]. The goal is to *reduce the dimensionality* of feature vectors: more specifically, to project high-dimensional feature vectors living in $\mathbb{R}^n$ into a lower dimensional space $\mathbb{R}^m$ (where $m \ll n$), while approximately preserving Euclidean distances (i.e. $\ell_2$ distances) with high probability. This dimensionality reduction enables a classifier to process vectors in $\mathbb{R}^m$, instead of vectors in $\mathbb{R}^n$. In this context, feature hashing was first introduced by Weinberger et. al [29] for document-based classification tasks such as email spam filtering. For such tasks, feature hashing yields a lower dimensional embedding of a high-dimensional feature vector derived from a bag-of-words model. Since then, feature hashing has become a mainstream approach [28], applied to numerous domains including ranking text documents [4], compressing neural networks [7], and protein sequence classification [5].

**Random Projections**

Dimensionality reduction schemes for feature vectors fit nicely into the random projection literature. In fact, the feature hashing scheme proposed by Weinberger et al. [29] boils down to uniformly drawing a random $m \times n$ matrix where each column contains *one* nonzero entry, equal to $-1$ or $1$.

The $\ell_2$-norm-preserving objective can be expressed mathematically as follows: for error $\epsilon > 0$ and failure probability $\delta$, the goal is to construct a probability distribution $\mathcal{A}$ over $m \times n$ real matrices

---

[*]I would like to thank Prof. Jelani Nelson for advising this project.

that satisfies the following condition for vectors $x \in \mathbb{R}^n$:

$$\mathbb{P}_{A \in \mathcal{A}}[(1 - \epsilon)\|x\|_2 \leq \|Ax\|_2 \leq (1 + \epsilon)\|x\|_2] > 1 - \delta. \tag{1}$$

The result underlying the random projection literature is the Johnson-Lindenstrauss lemma, which gives an upper bound on the dimension $m$ achievable by a probability distribution $\mathcal{A}$ satisfying (1):

**Lemma 1.1 (Johnson-Lindenstrauss [16])** For any $n \in \mathbb{N}$ and $\epsilon, \delta \in (0, 1)$, there exists a probability distribution $\mathcal{A}$ over $m \times n$ matrices, with $m = \Theta(\epsilon^{-2}\ln(1/\delta))$, that satisfies (1).

The optimality of the dimension $m$ achieved by Lemma 1.1 has been proven [17, 15].

To speed up projection time, it is useful to consider probability distributions over sparse matrices (i.e. matrices with a small number of nonzero entries per column). More specifically, for matrices with $s$ nonzero entries per column, the projection time for a vector $x$ goes down from $O(m\|x\|_0)$ to $O(s\|x\|_0)$, where $\|x\|_0$ is the number of nonzero entries of $x$. In this context, Kane and Nelson [19] constructed sparse JL distributions (which we define formally in Section 1.1), improving upon previous work [2, 22, 12]. Roughly speaking, a sparse JL distribution, as constructed in [19], boils down to drawing a random $m \times n$ matrix where each column contains exactly $s$ nonzero entries, each equal to $-1/\sqrt{s}$ or $1/\sqrt{s}$. Kane and Nelson show that sparse JL distributions achieve the same (optimal) dimension as Lemma 1.1, while also satisfying a sparsity property.

**Theorem 1.2 (Sparse JL [19])** For any $n \in \mathbb{N}$ and $\epsilon, \delta \in (0, 1)$, a sparse JL distribution $\mathcal{A}_{s,m,n}$ (defined formally in Section 1.1) over $m \times n$ matrices, with dimension $m = \Theta(\epsilon^{-2}\ln(1/\delta))$ and sparsity $s = \Theta(\epsilon^{-1}\ln(1/\delta))$, satisfies (1).

Sparse JL distributions are state-of-the-art sparse random projections, and achieve a sparsity that is nearly optimal when the dimension $m$ is $\Theta(\epsilon^{-2}\ln(1/\delta))$.[2] However, in practice, it can be necessary to utilize a lower sparsity $s$, since the projection time is linear in $s$. Resolving this issue, Cohen [8] extended the upper bound in Theorem 1.2 to show that sparse JL distributions can achieve a lower sparsity with an appropriate gain in dimension. He proved the following dimension-sparsity tradeoffs:

**Theorem 1.3 (Dimension-Sparsity Tradeoffs [8])** For any $n \in \mathbb{N}$ and $\epsilon, \delta \in (0, 1)$, a uniform sparse JL distribution $\mathcal{A}_{s,m,n}$ (defined formally in Section 1.1), with $s \leq \Theta(\epsilon^{-1}\ln(1/\delta))$ and $m \geq \min\left(2\epsilon^{-2}/\delta, \epsilon^{-2}\ln(1/\delta)e^{\Theta(\epsilon^{-1}\ln(1/\delta)/s)}\right)$, satisfies (1).

**Connection to Feature Hashing**

Sparse JL distributions have particularly close ties to feature hashing. In particular, the feature hashing scheme proposed by Weinberger et al. [29] can be viewed as a special case of sparse JL, namely with $s = 1$. Interestingly, in practice, feature hashing can do much better than theoretical results, such as Theorem 1.2 and Theorem 1.3, would indicate [13]. An explanation for this phenomenon is that the highest error terms in sparse JL stem from vectors with mass concentrated on a very small number of entries, while in practice, the mass on feature vectors may be spread out between many coordinates. This motivates studying the tradeoff space for vectors with low $\ell_\infty$-to-$\ell_2$ ratio.

More formally, take $S_v$ to be $\left\{x \in \mathbb{R}^n \mid \frac{\|x\|_\infty}{\|x\|_2} \leq v\right\}$, so that $S_1 = \mathbb{R}^n$ and $S_v \subsetneq S_w$ for $0 \leq v < w \leq 1$. Let $v(m, \epsilon, \delta, s)$ be the supremum over all $0 \leq v \leq 1$ such that a sparse JL distribution with sparsity $s$ and dimension $m$ satisfies (1) for each $x \in S_v$. (That is, $v(m, \epsilon, \delta, s)$ is the maximum $v \in [0, 1]$ such that for every $x \in \mathbb{R}^n$, if $\|x\|_\infty \leq v\|x\|_2$ then (1) holds.) For $s = 1$, a line of work [29, 12, 18, 10, 19] improved bounds on $v(m, \epsilon, \delta, 1)$, and was recently closed by Freksen et al. [13].

**Theorem 1.4 ([13])** For any $m \in \mathbb{N}$ and $\epsilon, \delta \in (0, 1)$, the function $v(m, \epsilon, \delta, 1)$ is equal to $f(m, \epsilon, \ln(1/\delta))$ where:

$$f(m, \epsilon, p) = \begin{cases} 1 & \text{if } m \geq 2\epsilon^{-2}e^p \\ \Theta\left(\sqrt{\epsilon}\min\left(\frac{\ln(\frac{m\epsilon}{p})}{p}, \frac{\sqrt{\ln(\frac{m\epsilon^2}{p})}}{\sqrt{p}}\right)\right) & \text{if } \Theta(\epsilon^{-2}p) \leq m < 2\epsilon^{-2}e^p \\ 0 & \text{if } m \leq \Theta(\epsilon^{-2}p). \end{cases}$$

**Generalizing to Sparse Random Projections with $s > 1$**

While Theorem 1.4 is restricted to the case of $s = 1$, dimensionality reduction schemes constructed using sparse random projections with sparsity $s > 1$ have been used in practice for projecting feature vectors. For example, sparse JL-like methods (with $s > 1$) have been used to project feature vectors in machine learning domains including visual tracking [27], face recognition [23], and recently in ELM [6]. Now, a variant of sparse JL is included in the Python sklearn library.[3]

In this context, it is natural to explore how constructions with $s > 1$ perform on feature vectors, by studying $v(m, \epsilon, \delta, s)$ for sparse JL with $s > 1$. In fact, a related question was considered by Weinberger et al. [29] for "multiple hashing," an alternate distribution over sparse matrices constructed by adding $s$ draws from $\mathcal{A}_{1,m,n}$ and scaling by $1/\sqrt{s}$. More specifically, they show that $v(m, \epsilon, \delta, s) \geq \min(1, \sqrt{s} \cdot v(m, \epsilon, \delta, 1))$ for multiple hashing. However, Kane and Nelson [19] later showed that multiple hashing has worse geometry-preserving properties than sparse JL: that is, multiple hashing requires a larger sparsity than sparse JL to satisfy (1).

Characterizing $v(m, \epsilon, \delta, s)$ for sparse JL distributions, which are state-of-the-art, remained an open problem. In this work, we settle how $v(m, \epsilon, \delta, s)$ behaves for sparse JL with a general sparsity $s > 1$, giving tight bounds. Our theoretical result shows that sparse JL with $s > 1$, even if $s$ is a small constant, can achieve significantly better norm-preservation properties for feature vectors than sparse JL with $s = 1$. Moreover, we empirically demonstrate this finding.

**Main Results**

We show the following tight bounds on $v(m, \epsilon, \delta, s)$ for a general sparsity $s$:

**Theorem 1.5** For any $s, m \in \mathbb{N}$ such that $s \leq m/e$, consider a uniform sparse JL distribution (defined in Section 1.1) with sparsity $s$ and dimension $m$.[4] If $\epsilon$ and $\delta$ are small enough[5], the function $v(m, \epsilon, \delta, s)$ is equal to $f'(m, \epsilon, \ln(1/\delta), s)$, where $f'(m, \epsilon, p, s)$ is[6]:

$$
\begin{cases}
1 & \text{if } m \geq \min\left(2\epsilon^{-2}e^p, \epsilon^{-2}pe^{\Theta\left(\max\left(1, \frac{p\epsilon^{-1}}{s}\right)\right)}\right) \\[2em]
\Theta\left(\sqrt{\epsilon s}\frac{\sqrt{\ln\left(\frac{m\epsilon^2}{p}\right)}}{\sqrt{p}}\right) & \text{else, if } \max\left(\Theta(\epsilon^{-2}p), s \cdot e^{\Theta\left(\max\left(1, \frac{p\epsilon^{-1}}{s}\right)\right)}\right) \leq m \leq \epsilon^{-2}e^{\Theta(p)} \\[2em]
\Theta\left(\sqrt{\epsilon s}\min\left(\frac{\ln\left(\frac{m\epsilon}{p}\right)}{p}, \frac{\sqrt{\ln\left(\frac{m\epsilon^2}{p}\right)}}{\sqrt{p}}\right)\right) & \text{else, if } \Theta(\epsilon^{-2}p) \leq m \leq \min\left(\epsilon^{-2}e^{\Theta(p)}, s \cdot e^{\Theta\left(\max\left(1, \frac{p\epsilon^{-1}}{s}\right)\right)}\right) \\[2em]
0 & \text{if } m \leq \Theta(\epsilon^{-2}p).
\end{cases}
$$

Our main result, Theorem 1.5, significantly generalizes Theorem 1.2, Theorem 1.3, and Theorem 1.4. Notice our bound in Theorem 1.5 has up to four regimes. In the first regime, which occurs when $m \geq \min(2\epsilon^{-2}/\delta, \epsilon^{-2}\ln(1/\delta)e^{\Theta(\max(1, \ln(1/\delta)\epsilon^{-1}/s))})$, Theorem 1.5 shows $v(m, \epsilon, \delta, s) = 1$, so (1) holds on the full space $\mathbb{R}^n$. Notice this boundary on $m$ occurs at the dimensionality-sparsity tradeoff in Theorem 1.3. In the last regime, which occurs when $m \leq \Theta(\epsilon^{-2}\ln(1/\delta))$, Theorem 1.5 shows that $v(m, \epsilon, \delta, s) = 0$, so there are vectors with arbitrarily small $\ell_\infty$-to-$\ell_2$ norm ratio where (1) does not hold. When $s \leq \Theta(\epsilon^{-1}\ln(1/\delta))$, Theorem 1.5 shows that up to two intermediate regimes exist. One of the regimes, $\Theta(\sqrt{\epsilon s}\min(\ln(\frac{m\epsilon}{p})/p, \sqrt{\ln(\frac{m\epsilon^2}{p})}/\sqrt{p}))$, matches the middle regime of $v(m, \epsilon, \delta, 1)$ in Theorem 1.4 with an extra factor of $\sqrt{s}$, much like the bound for multiple hashing in [29] that we mentioned previously. However, unlike the multiple hashing bound, Theorem 1.5 sometimes has another regime, $\Theta(\sqrt{\epsilon s}\sqrt{\ln(\frac{m\epsilon^2}{p})}/\sqrt{p})$, which does not arise for $s = 1$ (i.e. in Theorem 1.4).[7] Intuitively, we expect this additional regime for sparse JL with $s$ close to $\Theta(\epsilon^{-1}\ln(1/\delta))$:

at $s = \Theta(\epsilon^{-1} \ln(1/\delta))$ and $m = \Theta(\epsilon^{-2} \ln(1/\delta))$, Theorem 1.2 tells us $v(m, \epsilon, \delta, s) = 1$, but if $\epsilon$ is a constant, then the branch $\Theta(\sqrt{\epsilon s} \ln\left(\frac{m\epsilon}{p}\right)/p)$ yields $\Theta(1/\sqrt{\ln(1/\delta)})$, while the branch $\Theta(\sqrt{\epsilon s} \sqrt{\ln(\frac{m\epsilon^2}{p})}/\sqrt{p})$ yields $\Theta(1)$. Thus, it is natural that the first branch disappears for large $m$.

Our result elucidates that $v(m, \epsilon, \delta, s)$ increases approximately as $\sqrt{s}$, thus providing insight into how even small constant increases in sparsity can be useful in practice. Another consequence of our result is a lower bound on dimension-sparsity tradeoffs (Corollary A.1 in Appendix A) that essentially matches the upper bound in Theorem 1.3. Moreover, we require new techniques to prove Theorem 1.5, for reasons that we discuss further in Section 1.2.

We also empirically support our theoretical findings in Theorem 1.5. First, we illustrate with real-world datasets the potential benefits of using small constants $s > 1$ for sparse JL on feature vectors. We specifically show that $s = \{4, 8, 16\}$ consistently outperforms $s = 1$ in preserving the $\ell_2$ norm of each vector, and that there can be up to a *factor of ten* decrease in failure probability for $s = 8, 16$ in comparison to $s = 1$. Second, we use synthetic data to illustrate phase transitions and other trends in Theorem 1.5. More specifically, we empirically show that $v(m, \epsilon, \delta, s)$ is not smooth, and that the middle regime(s) of $v(m, \epsilon, \delta, s)$ increases with $s$.

## 1.1 Preliminaries

Let $\mathcal{A}_{s,m,n}$ be a **sparse JL distribution** if the entries of a matrix $A \in \mathcal{A}_{s,m,n}$ are generated as follows. Let $A_{r,i} = \eta_{r,i}\sigma_{r,i}/\sqrt{s}$ where $\{\sigma_{r,i}\}_{r\in[m],i\in[n]}$ and $\{\eta_{r,i}\}_{r\in[m],i\in[n]}$ are defined as follows:

- The families $\{\sigma_{r,i}\}_{r\in[m],i\in[n]}$ and $\{\eta_{r,i}\}_{r\in[m],i\in[n]}$ are independent from each other.
- The variables $\{\sigma_{r,i}\}_{r\in[m],i\in[n]}$ are i.i.d Rademachers ($\pm 1$ coin flips).
- The variables $\{\eta_{r,i}\}_{r\in[m],i\in[n]}$ are identically distributed Bernoullis ($\{0, 1\}$ random variables) with expectation $s/m$.
- The $\{\eta_{r,i}\}_{r\in[m],i\in[n]}$ are independent across columns but not independent within each column. For every column $1 \le i \le n$, it holds that $\sum_{r=1}^{m} \eta_{r,i} = s$. Moreover, the random variables are *negatively correlated*: for every subset $S \subseteq [m]$ and every column $1 \le i \le n$, it holds that $\mathbb{E}\left[\prod_{r\in S} \eta_{r,i}\right] \le \prod_{r\in S} \mathbb{E}[\eta_{r,i}]$.

A common special case is a **uniform sparse JL distribution**, generated as follows: for every $1 \le i \le n$, we *uniformly* choose exactly $s$ of these variables in $\{\eta_{r,i}\}_{r\in[m]}$ to be 1. When $s = 1$, every sparse JL distribution is a uniform sparse JL distribution, but for $s > 1$, this is not the case.

Another common special case is a **block sparse JL distribution**. This produces a different construction for $s > 1$. In this distribution, each column $1 \le i \le n$ is partitioned into $s$ blocks of $\lfloor \frac{m}{s} \rfloor$ consecutive rows. In each block in each column, the distribution of the variables $\{\eta_{r,i}\}$ is defined by uniformly choosing *exactly one* of these variables to be 1.[8]

## 1.2 Proof Techniques

We use the following notation. For any random variable $X$ and value $q \ge 1$, we call $\mathbb{E}[|X|^q]$ the $q$th *moment* of $X$, where $\mathbb{E}$ denotes the expectation. We use $\|X\|_q$ to denote the $q$-norm $(\mathbb{E}[|X|^q])^{1/q}$.

For every $[x_1, \ldots, x_n] \in \mathbb{R}^n$ such that $\|x\|_2 = 1$, we need to analyze tail bounds of an error term, which for the sparse JL construction is the following random variable:

$$\|Ax\|_2^2 - 1 = \frac{1}{s}\sum_{i \ne j}\sum_{r=1}^{m} \eta_{r,i}\eta_{r,j}\sigma_{r,i}\sigma_{r,j}x_i x_j =: R(x_1, \ldots, x_n).$$

An upper bound on the tail probability of $R(x_1, \ldots, x_n)$ is needed to prove the lower bound on $v(m, \epsilon, \delta, s)$ in Theorem 1.5, and a lower bound is needed to prove the upper bound on $v(m, \epsilon, \delta, s)$

in Theorem 1.5. It turns out that it suffices to tightly analyze the random variable moments $\mathbb{E}[(R(x_1, \ldots, x_n))^q]$. For the upper bound, we use Markov's inequality like in [13, 19, 3, 24], and for the lower bound, we use the Paley-Zygmund inequality like in [13]: Markov's inequality gives a tail upper bound from upper bounds on moments, and the Paley-Zygmund inequality gives a tail lower bound from upper and lower bounds on moments. Thus, the key ingredient of our analysis is a *tight bound* for $\|R(x_1, \ldots, x_n)\|_q$ on $S_v = \left\{ x \in \mathbb{R}^n \mid \frac{\|x\|_\infty}{\|x\|_2} \leq v \right\}$ at *each* threshold $v$ value.

While the moments of $R(x_1, \ldots, x_n)$ have been studied in previous analyses of sparse JL, we emphasize that it is not clear how to adapt these existing approaches to obtain a tight bound on every $S_v$. The moment bound that we require and obtain is far more general: the bounds in [19, 9] are limited to $\mathbb{R}^n = S_1$ and the bound in [13] is limited to $s = 1$.[9] The non-combinatorial approach in [9] for bounding $\|R(x_1, \ldots, x_n)\|_q$ on $\mathbb{R}^n = S_1$ also turns out to not be sufficiently precise on $S_v$, for reasons we discuss in Section 2.[10]

Thus, we require new tools for our moment bound. Our analysis provides a new perspective, inspired by the probability theory literature, that differs from the existing approaches in the JL literature. We believe our style of analysis is less brittle than combinatorial approaches [13, 19, 3, 24]: in this setting, once the sparsity $s = 1$ case is recovered, it becomes straightforward to generalize to other $s$ values. Moreover, our approach can yield greater precision than the existing non-combinatorial approaches [9, 8, 14], which is necessary for this setting. Thus, we believe that our *structural* approach to analyzing JL distributions could be of use in other settings.

In Section 2, we present an overview of our methods and the key technical lemmas to analyze $\|R(x_1, \ldots, x_n)\|_q$. We defer the proofs to the Appendix. In Section 3, we prove the tail bounds in Theorem 1.5 from these moment bounds. In Section 4, we empirically evaluate sparse JL.

## 2    Sketch of Bounding the Moments of $R(x_1, \ldots, x_n)$

Our approach takes advantage of the structure of $R(x_1, \ldots, x_n)$ as a quadratic form of Rademachers (i.e. $\sum_{t_1, t_2} a_{t_1, t_2} \sigma_{t_1} \sigma_{t_2}$) with random variable coefficients (i.e. where $a_{t_1, t_2}$ is itself a random variable). For the upper bound, we need to analyze $\|R(x_1, \ldots, x_n)\|_q$ for general vectors $[x_1, \ldots, x_n]$. For the lower bound, we only need to show $\|R(x_1, \ldots, x_n)\|_q$ is large for single vector in each $S_v$, and we show we can select the vector in the $\ell_2$-unit ball with $1/v^2$ nonzero entries, all equal to $v$. For ease of notation, we denote this vector by $[v, \ldots, v, 0, \ldots, 0]$ for the remainder of the paper.

We analyze $\|R(x_1, \ldots, x_n)\|_q$ using general moment bounds for Rademacher linear and quadratic forms. Though Cohen, Jayram, and Nelson [9] also view $R(x_1, \ldots, x_n)$ as a quadratic form, we show in the supplementary material that their approach of bounding the Rademachers by gaussians is not sufficiently precise for our setting.[11]

In our approach, we make use of stronger moment bounds for Rademacher linear and quadratic forms, some of which are known to the probability theory community through Latała's work in [21, 20] and some of which are new adaptions tailored to the constraints arising in our setting. More specifically, Latała's bounds [21, 20] target the setting where the coefficients are scalars. In our setting, however, the coefficients are themselves random variables, and we need bounds that are *tractable* to analyze in this setting, which involves creating new bounds to handle some cases.

Our strategy for bounding $\|R(x_1, \ldots, x_n)\|_q$ is to break down into rows. We define

$$Z_r(x_1, \ldots, x_n) := \sum_{1 \leq i \neq j \leq n} \eta_{r,i} \eta_{r,j} \sigma_{r,i} \sigma_{r,j} x_i x_j$$

so that $R(x_1, \ldots, x_n) = \frac{1}{s} \sum_{r=1}^{m} Z_r(x_1, \ldots, x_n)$. We analyze the moments of $Z_r(x_1, \ldots, x_n)$, and then combine these bounds to obtain moment bounds for $R(x_1, \ldots, x_n)$. In our bounds, we use the notation $f \lesssim g$ (resp. $f \gtrsim g$) to denote $f \leq Cg$ (resp. $f \geq Cg$) for some constant $C$.

## 2.1 Bounding $\|Z_r(x_1, \ldots, x_n)\|_q$

We show the following bounds on $\|Z_r(x_1, \ldots, x_n)\|_q$. For the lower bound, as we discussed before, it suffices to bound $\|Z_r(v, \ldots, v, 0, \ldots, 0)\|_q$. For the upper bound, we need to bound $\|Z_r(x_1, \ldots, x_n)\|_q$ for general vectors as a function of the $\ell_\infty$-to-$\ell_2$ norm ratio.

**Lemma 2.1** Let $\mathcal{A}_{s,m,n}$ be a sparse JL distribution such that $s \leq m/e$. Suppose that $x = [x_1, \ldots, x_n]$ satisfies $\|x\|_\infty \leq v$ and $\|x\|_2 = 1$. If $T$ is even, then:

$$\|Z_r(x_1, \ldots, x_n)\|_T \lesssim \begin{cases} \frac{Ts}{m}, & \text{for } T = 2, 3 \leq T \leq \frac{se}{mv^2} \\ \min\left(\frac{T^2 v^2}{\ln(mTv^2/s)^2}, \frac{T}{\ln(m/s)}\right) & \text{for } T \geq 3, T \geq \frac{se}{mv^2}, \ln(Tmv^2/s) \leq T \\ v^2 \left(\frac{s}{mTv^2}\right)^{2/T}, & \text{for } T \geq 3, T \geq \frac{se}{mv^2}, \ln(Tmv^2/s) > T. \end{cases}$$

**Lemma 2.2** Let $\mathcal{A}_{s,m,n}$ be a sparse JL distribution. Suppose $\frac{1}{v^2}$ and $T$ are even integers. Then, $\|Z_r(v, \ldots, v, 0, \ldots, 0)\|_2 \gtrsim \frac{s}{m}$. Moreover, if $s \leq m/e$ and $T \geq \frac{se}{mv^2}$, then $\left\|Z_r(v, \ldots, v, 0, \ldots, 0) I_{\sum_{i=1}^{1/v^2} \eta_{1,i}=2}\right\|_T \gtrsim v^2 \left(\frac{s}{mTv^2}\right)^{2/T}$ and

$$\|Z_r(v, \ldots, v, 0, \ldots, 0)\|_T \gtrsim \begin{cases} \frac{T^2 v^2}{\ln^2(mv^2 T/s)} & \text{for } 1 \leq \ln(mv^2 T/s) \leq T, v \leq \frac{\sqrt{\ln(m/s)}}{\sqrt{T}} \\ v^2 \left(\frac{s}{mTv^2}\right)^{2/T} & \text{for } \ln(mv^2 T/s) > T. \end{cases}$$

We now sketch our methods to prove Lemma 2.1 and Lemma 2.2. For the lower bound (Lemma 2.2), we can view $Z_r(v, \ldots, v, 0, \ldots, 0)$ as a quadratic form $\sum_{t_1, t_2} a_{t_1, t_2} \sigma_{t_1} \sigma_{t_2}$ where $(a_{t_1, t_2})_{t_1, t_2 \in [mn]}$ is an appropriately defined block-diagonal $mn$ dimensional matrix. We can write $\mathbb{E}_{\sigma, \eta}[(Z_r(v, \ldots, v, 0, \ldots, 0))^q]$ as $\mathbb{E}_\eta [\mathbb{E}_\sigma[(Z_r(v, \ldots, v, 0, \ldots, 0))^q]]$: for *fixed* $\eta_{r,i}$ values, the coefficients are scalars. We make use of Latała's tight bound on Rademacher quadratic forms with scalar coefficients [21] to analyze $\mathbb{E}_\sigma[(Z_r(v, \ldots, v, 0, \ldots, 0))^q]$ as a function of the $\eta_{r,i}$. Then, we handle the randomness of the $\eta_{r,i}$ by taking an expectation of the resulting bound on $\mathbb{E}_\sigma[(Z_r(v, \ldots, v, 0, \ldots, 0))^q]$ over the $\eta_{r,i}$ values to obtain a bound on $\|Z_r(v, \ldots, v, 0, \ldots, 0)\|_q$.

For the upper bound (Lemma 2.1), since Latała's bound [21] is tight for scalar quadratic forms, the natural approach would be to use it to upper bound $\mathbb{E}_\sigma[(Z_r(x_1, \ldots, x_n))^q]$ for general vectors. However, when the vector is not of the form $[v, \ldots, v, 0, \ldots, 0]$, the asymmetry makes the resulting bound intractable to simplify. Specifically, there is a term, which can be viewed as a generalization of an operator norm to an $\ell_2$ ball cut out by $\ell_\infty$ hyperplanes, that becomes problematic when taking an expectation over the $\eta_{r,i}$ to obtain a bound on $\mathbb{E}_{\sigma, \eta}[(Z_r(x_1, \ldots, x_n))^q]$. Thus, we construct simpler estimates that avoid these complications while remaining sufficiently precise for our setting. These estimates take advantage of the structure of $Z_r(x_1, \ldots, x_n)$ and enable us to show Lemma 2.1.

## 2.2 Obtaining bounds on $\|R(x_1, \ldots, x_n)\|_q$

Now, we use Lemma 2.1 and Lemma 2.2 to show the following bounds on $\|R(x_1, \ldots, x_n)\|_q$:

**Lemma 2.3** Suppose $\mathcal{A}_{s,m,n}$ is a sparse JL distribution such that $s \leq m/e$, and let $x = [x_1, \ldots, x_n]$ be such that $\|x\|_2 = 1$. Then, $\|R(x_1, \ldots, x_n)\|_2 \leq \frac{\sqrt{2}}{\sqrt{m}}$. Now, suppose that $2 < q \leq m$ is an even integer and $\|x\|_\infty \leq v$. If $\frac{se}{mv^2} \geq q$, then $\|R(x_1, \ldots, x_n)\|_q \lesssim \frac{\sqrt{q}}{\sqrt{m}}$. If $\frac{se}{mv^2} < q$ and if there exists a constant $C_2 \geq 1$ such that $C_2 q^3 mv^4 \geq s^2$, then $\|R(x_1, \ldots, x_n)\|_q \lesssim g$ where $g$ is:

$$\begin{cases} \max\left(\frac{\sqrt{q}}{\sqrt{m}}, \frac{C_2^{1/3} q^2 v^2}{s \ln^2(qmv^2/s)}\right) & \text{if } \ln(\frac{qmv^4}{s^2}) \leq 2, \ln(\frac{qmv^2}{s}) \leq q \\ \frac{\sqrt{q}}{\sqrt{m}} & \text{if } \ln(\frac{qmv^4}{s^2}) \leq 2, \ln(\frac{qmv^2}{s}) > q \\ \max\left(\frac{\sqrt{q}}{\sqrt{m}}, \frac{qv^2}{s \ln(qmv^4/s^2)}, \min\left(\frac{C_2^{1/3} q^2 v^2}{s \ln^2(qmv^2/s)}, \frac{q}{s \ln(m/s)}\right)\right) & \text{if } \ln(\frac{qmv^4}{s^2}) > 2, \ln(\frac{qmv^2}{s}) \leq q \\ \max\left(\frac{\sqrt{q}}{\sqrt{m}}, \frac{qv^2}{s \ln(qmv^4/s^2)}\right) & \text{if } \ln(\frac{qmv^4}{s^2}) > 2, \ln(\frac{qmv^2}{s}) > q. \end{cases}$$

**Lemma 2.4** Suppose $\mathcal{A}_{s,m,n}$ is a uniform sparse JL distribution. Let $q$ be a power of 2, and suppose that $0 < v \leq 0.5$ and $\frac{1}{v^2}$ is an even integer. If $qv^2 \leq s$, then $\|R(v, \ldots, v, 0, \ldots, 0)\|_q \gtrsim \frac{\sqrt{q}}{\sqrt{m}}$. If $m \geq q$, $2 \leq \ln(qmv^4/s^2) \leq q$, $2qv^2 \leq 0.5s\ln(qmv^4/s^2)$, and $s \leq m/e$, then $\|R(v, \ldots, v, 0, \ldots, 0)\|_q \gtrsim \frac{qv^2}{s\ln(qmv^4/s^2)}$. If $s \leq m/e$, $v \leq \frac{\sqrt{\ln(m/s)}}{\sqrt{q}}$, and $1 \leq \ln(qmv^2/s) \leq q$, then $\|R(v, \ldots, v, 0 \ldots, 0)\|_q \gtrsim \frac{q^2v^2}{s\ln^2(qmv^2/s)}$.

We now sketch how to prove bounds on $\|R(x_1, \ldots, x_n)\|_q$ using bounds on $\|Z_r(x_1, \ldots, x_n)\|_T$. To show Lemma 2.3, we show that making the row terms $Z_r(x_1, \ldots, x_n)$ independent does not decrease $\|R(x_1, \ldots, x_n)\|_q$, and then we apply a general result from [20] for moments of sums of i.i.d symmetric random variables. For Lemma 2.4, handling the correlations between the row terms $Z_r(x_1, \ldots, x_n)$ requires more care. We show that the negative correlations induced by having exactly $s$ nonzero entries per column do not lead to significant loss, and then stitch together $\|R(v, \ldots, v, 0, \ldots, 0)\|_q$ using the moments of $Z_r(v, \ldots, v, 0, \ldots, 0)$ that contribute the most.

## 3 Proof of Main Result from Moment Bounds

We now sketch how to prove Theorem 1.5, using Lemma 2.3 and Lemma 2.4. First, we simplify these bounds at the target parameters to obtain the following:

**Lemma 3.1** Let $\mathcal{A}_{s,m,n}$ be a sparse JL distribution, and suppose $\epsilon$ and $\delta$ are small enough, $s \leq m/e$, $\Theta(\epsilon^{-2}\ln(1/\delta)) \leq m < 2\epsilon^{-2}/\delta$, $v \leq f'(m, \epsilon, \ln(1/\delta), s)$, and $p = \Theta(\ln(1/\delta))$ is even. If $x = [x_1, \ldots, x_n]$ satisfies $\|x\|_\infty \leq v$ and $\|x\|_2 = 1$, then $\|R(x_1, \ldots, x_n)\|_p \leq \frac{\epsilon}{2}$.

**Lemma 3.2** There is a universal constant $D$ satisfying the following property. Let $\mathcal{A}_{s,m,n}$ be a uniform sparse JL distribution, and suppose $\epsilon, \delta$ are small enough, $s \leq m/e$, $f'(m, \epsilon, \ln(1/\delta), s) \leq 0.5$, and $q$ is an even integer such that $q = \min(m/2, \Theta(\ln(1/\delta)))$. For each $\psi > 0$, there exists $v \leq f'(m, \epsilon, \ln(1/\delta), s) + \psi$, such that $\|R(v, \ldots, v, 0, \ldots, 0)\|_q \geq 2\epsilon$ and $\frac{\|R(v,\ldots,v,0,\ldots,0)\|_q}{\|R(v,\ldots,v,0,\ldots,0)\|_{2q}} \geq D$.

Now, we use Lemma 3.1 and Lemma 3.2 to prove Theorem 1.5.

*Proof of Theorem* 1.5. Since the maps in $\mathcal{A}_{s,m,n}$ are linear, it suffices to consider unit vectors $x$. First, we prove the lower bound on $v(m, \epsilon, \delta, s)$. To handle $m \geq 2\epsilon^{-2}/\delta$, we take $q = 2$ in Lemma 3.1 and apply Chebyshev's inequality. Otherwise, we take $p = \ln(1/\delta)$ (approximately) and apply Lemma 3.1 and Markov's inequality. We see that $\mathbb{P}[|\|Ax\|_2^2 - 1| \geq \epsilon]$ can be expressed as:

$$\mathbb{P}[|R(x_1, \ldots, x_n)| \geq \epsilon] = \mathbb{P}[R(x_1, \ldots, x_n)^p \geq \epsilon^p] \leq \epsilon^{-p}\mathbb{E}[R(x_1, \ldots, x_n)]^p \leq \delta.$$

Thus, condition (1) is satisfied for $x \in S_v$ when $v \leq f'(m, \epsilon, \ln(1/\delta), s)$ as desired.

Now, we prove the upper bound on $v(m, \epsilon, \delta, s)$. We need to lower bound the tail probability of $R(v, \ldots, v, 0, \ldots, 0)$, and to do this, we use the Paley-Zygmund inequality applied to $q$th moments. Let $D$ be defined as in Lemma 3.2, and take $q = \min(m/2, \frac{\ln(1/\delta)-2}{-2\ln(D)})$. By the Paley-Zygmund inequality and Lemma 3.2, there exists $v \leq f'(m, \epsilon, \ln(1/\delta), s) + \psi$ such that:

$$\mathbb{P}[|R(v, v, \ldots, v, 0, \ldots, 0)| > \epsilon] \geq 0.25 \left(\frac{\|R(v, v, \ldots, v, 0, \ldots, 0)\|_q}{\|R(v, v, \ldots, v, 0, \ldots, 0)\|_{2q}}\right)^{2q} \geq 0.25D^{2q} > \delta.$$

Thus, it follows that $\sup_{x \in S_{f'(m,\epsilon,\ln(1/\delta),s)+\psi}, \|x\|_2=1} \mathbb{P}[|\|Ax\|_2^2 - 1| > \epsilon] > \delta$ as desired. $\square$

## 4 Empirical Evaluation

Recall that for sparse JL distributions with sparsity $s$, the projection time for an input vector $x$ is $O(s\|x\|_0)$, where $\|x\|_0$ is the number of nonzero entries in $x$. Since this grows linearly in $s$, in order to minimize the impact on projection time, we restrict to small constant $s$ values (i.e. $1 \leq s \leq 16$). In Section 4.1, we demonstrate on real-world data the benefits of using $s > 1$. In Section 4.2, we illustrate trends in our theoretical bounds on synthetic data. Additional graphs can be found in Appendix I. For all experiments, we use a block sparse JL distribution to demonstrate that our theoretical upper bounds also empirically generalize to non-uniform sparse JL distributions.

## 4.1 Real-World Datasets

We considered two bag-of-words datasets: the News20 dataset [1] (based on newsgroup documents), and the Enron email dataset [26] (based on e-mails from the senior management of Enron).[12] Both datasets were pre-processed with the standard `tf-idf` preprocessing. In this experiment, we evaluated how well sparse JL preserves the $\ell_2$ norms of the vectors in the dataset. An interesting direction for future work would be to empirically evaluate how well sparse JL preserves other aspects of the geometry of real-world data sets, such as the $\ell_2$ distances between pairs of vectors.

In our experiment, we estimated the failure probability $\hat{\delta}(s, m, \epsilon)$ for each dataset as follows. Let $D$ be the number of vectors in the dataset, and let $n$ be the dimension ($n = 101631$, $D = 11314$ for News20; $n = 28102$, $D = 39861$ for Enron). We drew a matrix $M \sim \mathcal{A}_{s,m,n}$ from a block sparse JL distribution. Then, we computed $\frac{\|Mx\|_2}{\|x\|_2}$ for each vector $x$ in the dataset, and used these values to compute an estimate $\hat{\delta}(s, m, \epsilon) = \frac{\text{number of vectors } x \text{ such that } \frac{\|Mx\|_2}{\|x\|_2} \notin 1 \pm \epsilon}{D}$. We ran 100 trials to produce 100 estimates $\hat{\delta}(s, m, \epsilon)$.

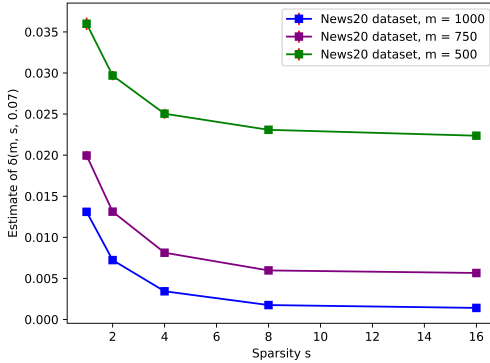

Figure 1: News20: $\hat{\delta}(m, s, 0.07)$ v. $s$

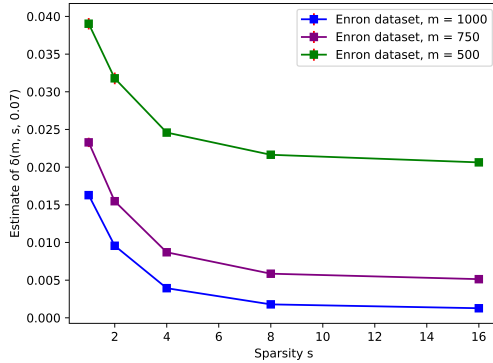

Figure 2: Enron: $\hat{\delta}(m, s, 0.07)$ vs. $s$

Figure 1 and Figure 2 show the mean and error bars (3 standard errors of the mean) of $\hat{\delta}(s, m, \epsilon)$ at $\epsilon = 0.07$. We consider $s \in \{1, 2, 4, 8, 16\}$, and choose $m$ values so that $0.01 \leq \hat{\delta}(1, m, \epsilon) \leq 0.04$.

All of the plots show that $s \in \{2, 4, 8, 16\}$ achieves a lower failure probability than $s = 1$, with the differences most pronounced when $m$ is larger. In fact, at $m = 1000$, there is a *factor of four* decrease in $\delta$ between $s = 1$ and $s = 4$, and a *factor of ten* decrease between $s = 1$ and $s = 8, 16$. We note that in plots in the Appendix, there is a slight increase between $s = 8$ and $s = 16$ at some $\epsilon, \delta, m$ values (see Appendix I for a discussion of this non-monotonicity in $s$); however $s > 1$ still consistently beats $s = 1$. Thus, these findings demonstrate the potential benefits of using small constants $s > 1$ in sparse JL in practice, which aligns with our theoretical results.

## 4.2 Synthetic Datasets

We used synthetic data to illustrate the phase transitions in our bounds on $v(m, \epsilon, \delta, s)$ in Theorem 1.5 for a block sparse JL distribution. For several choices of $s, m, \epsilon, \delta$, we computed an estimate $\hat{v}(m, \epsilon, \delta, s)$ of $v(m, \epsilon, \delta, s)$ as follows. Our experiment borrowed aspects of the experimental design in [13]. Our synthetic data consisted of binary vectors (i.e. vectors whose entries are in $\{0, 1\}$). The binary vectors were defined by a set $W$ of values exponentially spread between 0.03 and $1^{13}$: for each $w \in W$, we constructed a binary vector $x^w$ where the first $1/w^2$ entries are nonzero, and computed an estimate $\hat{\delta}(s, m, \epsilon, w)$ of the failure probability of the block sparse JL distribution on the specific vector $x^w$ (i.e. $\mathbb{P}_{A \in \mathcal{A}_{s,m,1/w^2}}[\|Ax^w\|_2 \notin (1 \pm \epsilon) \|x^w\|_2]$). We computed each $\hat{\delta}(s, m, \epsilon, w)$ using 100,000 samples from a block sparse JL distribution, as follows. In each sample, we independently

drew a matrix $M \sim \mathcal{A}_{s,m,1/w^2}$ and computed the ratio $\frac{\|Mx^w\|_2}{\|x^w\|_2}$. Then, we took $\hat{\delta}(s, m, \epsilon, w) :=$ (number of samples where $\frac{\|Mx^w\|_2}{\|x^w\|_2} \notin 1 \pm \epsilon)/T$. Finally, we used the estimates $\hat{\delta}(s, m, \epsilon, w)$ to obtain the estimate $\hat{v}(m, \epsilon, \delta, s) = \max \left\{ v \in W \mid \hat{\delta}(s, m, \epsilon, w) < \delta \text{ for all } w \in W \text{ where } w \leq v \right\}$.

Why does this procedure estimate $v(m, \epsilon, \delta, s)$? With enough samples, $\hat{\delta}(s, m, \epsilon, w) \to \mathbb{P}_{A \in \mathcal{A}_{s,m,1/w^2}}[\|Ax^w\|_2 \notin (1 \pm \epsilon) \|x^w\|_2]$.[14] As a result, if $x^w$ is a "violating" vector, i.e. $\hat{\delta}(s, m, \epsilon, w) \geq \delta$, then likely $\mathbb{P}_{A \in \mathcal{A}_{s,m,n}}[\|Ax^w\|_2 \notin (1 \pm \epsilon) \|x^w\|_2] \geq \delta$, and so $\hat{v}(m, \epsilon, \delta, s) \geq v(m, \epsilon, \delta, s)$. For the other direction, we use that in the proof of Theorem 1.5, we show that asymptotically, if a "violating" vector (i.e. $x$ s.t. $\mathbb{P}_{A \in \mathcal{A}_{s,m,n}}[\|Ax\|_2 \notin (1 \pm \epsilon) \|x\|_2] \geq \delta$) exists in $S_v$, then there's a "violating" vector of the form $x^w$ for some $w \leq \Theta(v)$. Thus, the estimate $\hat{v}(m, \epsilon, \delta, s) = \Theta(v(m, \epsilon, \delta, s))$ as $T \to \infty$ and as precision in $W$ goes to $\infty$.

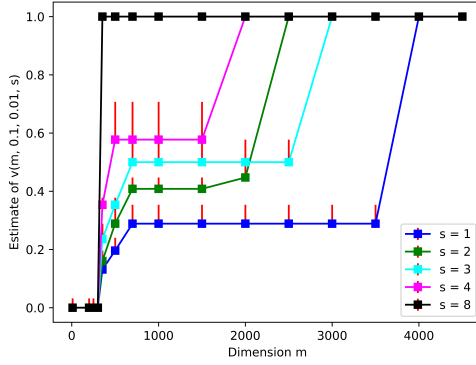 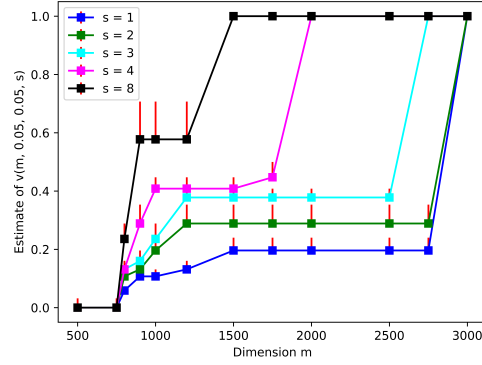

Figure 3: Phase transitions of $\hat{v}(m, 0.1, 0.01, s)$ Figure 4: Phase transitions of $\hat{v}(m, 0.05, 0.05, s)$

Figure 3 and Figure 4 show $\hat{v}(m, \epsilon, \delta, s)$ as a function of dimension $m$ for $s \in \{1, 2, 3, 4, 8\}$ for two settings of $\epsilon$ and $\delta$. The error-bars are based on the distance to the next highest $v$ value in $W$.

Our first observation is that for each set of $s, \epsilon, \delta$ values considered, the curve $\hat{v}(m, \epsilon, \delta, s)$ has "sharp" changes as a function of $m$. More specifically, $\hat{v}(m, \epsilon, \delta, s)$ is 0 at small $m$, then there is a phase transition to a nonzero value, then an increase to a higher value, then an interval where the value appears "flat", and lastly a second phase transition to 1. The first phase transition is shared between $s$ values, but the second phase transition occurs at different dimensions $m$ (but is within a factor of 3 between $s$ values). Here, the first phase transition likely corresponds to $\Theta(\epsilon^{-2} \ln(1/\delta))$ and the second phase transition likely corresponds to $\min \left( \epsilon^{-2} e^{\Theta(\ln(1/\delta))}, \epsilon^{-2} \ln(1/\delta) e^{\Theta(\ln(1/\delta)\epsilon^{-1}/s)} \right)$.

Our second observation is that as $s$ increases, the "flat" part occurs at a higher y-coordinate. Here, the increase in the "flat" y-coordinate as a function of $s$ corresponds to the $\sqrt{s}$ term in $v(m, \epsilon, \delta, s)$. Technically, according to Theorem 1.5, the "flat" parts should be increasing in $m$ at a slow rate: the empirical "flatness" likely arises since $W$ is a finite set in the experiments.

Our third observation is that $s > 1$ generally outperforms $s = 1$ as Theorem 1.5 suggests: that is, $s > 1$ generally attains a higher $\hat{v}(m, \epsilon, \delta, s)$ value than $s = 1$. We note at large $m$ values (where $\hat{v}(m, \epsilon, \delta, s)$ is close to 1), lower $s$ settings sometimes attain a higher $\hat{v}(m, \epsilon, \delta, s)$ than higher $s$ settings (e.g. the second phase transition doesn't quite occur in decreasing order of $s$ in Figure 3): see Appendix I for a discussion of this non-monotonicity in $s$.[15] Nonetheless, in practice, it's unlikely to select such a large dimension $m$, since the $\ell_\infty$-to-$\ell_2$ guarantees of smaller $m$ are likely sufficient. Hence, a greater sparsity generally leads to a better $\hat{v}(m, \epsilon, \delta, s)$ value, thus aligning with our theoretical findings.

## Footnotes

[2]Nelson and Nguyen [25] showed that *any* distribution satisfying (1) requires sparsity $\Omega(\epsilon^{-1}\ln(1/\delta)/\ln(1/\epsilon))$ when the dimension $m$ is $\Theta(\epsilon^{-2}\ln(1/\delta))$. Kane and Nelson [19] also showed that the analysis of sparse JL distributions in Theorem 1.2 is tight at $m = \Theta(\epsilon^{-2}\ln(1/\delta))$.

[3]See https://scikit-learn.org/stable/modules/random_projection.html.

[4]We prove the lower bound on $v(m, \epsilon, \delta, s)$ in Theorem 1.5 for *any* sparse JL distribution.

[5]By "small enough", we mean the condition that $\epsilon, \delta \in (0, C')$ for some positive constant $C'$.

[6]Notice that the function $f'(m, \epsilon, p, s)$ is not defined for certain "constant-factor" intervals between the boundaries of regimes (e.g. $C_1\epsilon^{-2}p \leq m \leq C_2\epsilon^{-2}p$). See Appendix A for a discussion.

[7]This regime does not arise for $s = 1$, since $e^{\Theta(p\epsilon^{-1})} \geq \epsilon^{-2}e^{\Theta(p)}$ for sufficiently small $\epsilon$.

[8]Our lower bound in Theorem 1.5 applies to this distribution, though our upper bound does not. An interesting direction for future work would be to generalize the upper bound to this distribution.

[9]As described in [13], even for the case for $s = 1$, the approach in [19] cannot be directly generalized to recover Theorem 1.4. Moreover, the approach in [13], though more precise for $s = 1$, is highly tailored to $s = 1$, and it is not clear how to generalize it to $s > 1$.

[10]In predecessor work [14], we give a non-combinatorial approach similar to [9] for a sign-consistent variant of the JL distribution. Moreover, a different non-combinatorial approach for subspace embeddings is given in [8]. However, these approaches both suffer from issues in this setting that are similar to [9].

[11]We actually made a similar conceptual point for a different JL distribution in our predecessor work [14], but the alternate bound that we produce there also suffers from precision issues in this setting.

[12]Note that the News20 dataset is used in [10], and the Enron dataset is from the same collection as the dataset used in [13], but contains a larger number of documents.

[13]We took $W = \left\{ w \mid w^{-2} \in \{986, 657, 438, 292, 195, 130, 87, 58, 39, 26, 18, 12, 9, 8, 7, 6, 5, 4, 3, 2, 1\} \right\}$.

[14]With 100,000 samples, running our procedure twice yielded the same $\hat{v}(m, \epsilon, \delta, s)$ values both times.

[15]In Appendix I, we also show more examples where at large $m$ values, lower $s$ settings attain a higher $\hat{v}(m, \epsilon, \delta, s)$ than higher $s$ settings.

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
