[Supplementary Material]

# Appendix for "Understanding Sparse JL for Feature Hashing"

November 25, 2019

In Appendix A, we prove our corollary regarding dimension-sparsity tradeoffs and discuss some of the subtleties of Theorem 1.5. In Appendix B, we show that the Hanson-Wright bound is too loose to prove Theorem 1.5. In Appendix C, we state and prove useful moment bounds that we use throughout the analysis. In Appendix D, we prove our moment bounds for $Z_r(x_1, \ldots, x_n)$ in Lemma 2.1 and Lemma 2.2. In Appendix E, we prove our moment bounds for $R(x_1, \ldots, x_n)$ in Lemma 2.3 and Lemma 2.4. In Appendix F, we prove auxiliary lemmas needed in the proof of Lemma 2.3. In Appendix G, we prove auxiliary lemmas needed in the proof of Lemma 2.4. In Appendix H, we prove our simplified moment bounds for $R(x_1, \ldots, x_n)$ in Lemma 3.1 and Lemma 3.2. In Appendix I, we provide additional experimental results on real-world and synthetic datasets as well as additional discussion.

## A  Discussion of Theoretical Results

We discuss some of the subtleties of Theorem 1.5. When $m \geq \min(2\epsilon^{-2}e^p, \epsilon^{-2}pe^{\Theta(\max(1, p\epsilon^{-1}/s))})$, where $p = \ln(1/\delta)$, we show that $v(m, \epsilon, \delta, s) = 1$, which means that the norm-preserving condition holds on the full space. This generalizes Cohen's bound [8] to a slightly more general family of sparse JL distributions, as we discuss below. When $m \leq \Theta(\epsilon^{-2} \ln(1/\delta))$, we show that $v(m, \epsilon, \delta, s) = 0$. For the remaining regimes, $\sqrt{\epsilon s}\sqrt{\ln(\frac{m\epsilon^2}{p})}/\sqrt{p}$ and $\sqrt{\epsilon s} \min\left(\ln(\frac{m\epsilon}{p})/p, \sqrt{\ln(\frac{m\epsilon^2}{p})}/\sqrt{p}\right)$, our upper and lower bounds on $v(m, \epsilon, \delta, s)$ match up to constant factors.

In terms of the boundaries between regimes, we emphasize that in Theorem 1.5, the function $f'(m, \epsilon, \delta, s)$ may not be defined for certain intervals between the boundaries of regimes, since there may be different absolute constants in different boundaries. More specifically, these intervals are $C_1\epsilon^{-2}p \leq m \leq C_2\epsilon^{-2}p$, $\epsilon^{-2}e^{C_1 p} \leq m \leq 2\epsilon^{-2}e^p$, and $s \cdot e^{C_1 \max(1, p\epsilon^{-1}/s)} \leq m \leq s \cdot e^{C_2 \max(1, p\epsilon^{-1}/s)}$. These gaps arise because the boundaries between the regimes on our upper and lower bounds on $v(m, \epsilon, \delta, s)$ can have different absolute constants, so we don't have precise control on $v(m, \epsilon, \delta, s)$ in these gaps. Nonetheless, the gaps only span a constant factor range on the exponent in the dimension $m$.

We now state the dimension-sparsity tradeoffs that follow from our bounds:

**Corollary A.1** Suppose that $\epsilon$ and $\delta$ are sufficiently small and $s \leq m/e$. If $\mathcal{A}_{s,m,n}$ is any sparse JL distribution, then $v(m, \epsilon, \delta, s) = 1$ when $m \geq \min\left(2\epsilon^{-2}/\delta, \epsilon^{-2}\ln(1/\delta)e^{\Theta(\max(1, \ln(1/\delta)\epsilon^{-1}/s))}\right)$. If $\mathcal{A}_{s,m,n}$ is a uniform sparse JL distribution, then $v(m, \epsilon, \delta, s) \leq 1/2$ when $m \leq \min\left(\epsilon^{-2}e^{\Theta(\ln(1/\delta))}, \epsilon^{-2}\ln(1/\delta)e^{\Theta(\max(1, \ln(1/\delta)\epsilon^{-1}/s))}\right)$, apart from a constant-factor interval $C_1\epsilon^{-2}\ln(1/\delta) \leq m \leq C_2\epsilon^{-2}\ln(1/\delta)$ where we do not have a bound on the behavior of sparse JL.

*Proof of Corollary A.1.* The first statement follows from the fact the lower bound in Theorem 1.5 holds for any sparse JL distribution. For the upper bound, we also use Theorem 1.5. Let's set $C_v\sqrt{\epsilon s}\frac{\sqrt{\ln(m\epsilon^2)}}{\sqrt{p}} = \frac{1}{2}$, where $C_v$ is the implicit constant in the upper bound. This solves to $m = \epsilon^{-2}pe^{\frac{C_L p\epsilon^{-1}}{s}}$ for some constant $C_L$ as desired. We also have the condition that $m \leq \epsilon^{-2}e^{\Theta(\ln(1/\delta))}$ for this regime to be reached. We can obtain the max with 1 on the exponent, by using that $v(m, \epsilon, \delta, s) = 0$ when $m \leq \Theta(\epsilon^{-2}\ln(1/\delta))$. To avoid having a gap when $m = s \cdot e^{\Theta(\max(1, \ln(1/\delta)\epsilon^{-1}/s))}$, we implicitly use that our lower bound actually doesn't have a gap between these regimes (though there may be a gap in the boundary between the lower bound and upper bound). Thus, we only have to keep the gap $C_1\epsilon^{-2}\ln(1/\delta) \leq m \leq C_2\epsilon^{-2}\ln(1/\delta)$ where we do not have a lower bound.  □

Notice that the upper and lower bounds in Corollary $A.1$ also match up to constant factors on the exponent in the dimension $m$.

## B  Hanson-Wright is too loose even for $s = 1$

Though Cohen, Jayram, and Nelson [9] also view $R(x_1, \ldots, x_n)$ as a quadratic form, we show that their approach is not sufficiently precise for our setting. They upper bound the moments of $R(x_1, \ldots, x_n)$ by the gaussian case through considering:

$$\tilde{R}(x_1, \ldots, x_n) = \frac{1}{s} \sum_{r=1}^{m} \sum_{i \neq j} \eta_{r,i} \eta_{r,j} g_{r,i} g_{r,j} x_i x_j$$

where the $g_{r,i}$ are i.i.d standard gaussians. They use the fact Rademachers are subgaussian to conclude that $\|R(x_1, \ldots, x_n)\|_q \leq \left\| \tilde{R}(x_1, \ldots, x_n) \right\|_q$. In order to obtain upper bounds on $\left\| \tilde{R}(x_1, \ldots, x_n) \right\|_q$, they use the Hanson-Wright bound, a tight bound on moments of gaussian quadratic forms. However, we need different technical tools for two reasons.

1. First, in order to upper bound $v(m, \epsilon, \delta, s)$, we need to *lower bound* $\|R(x_1, \ldots, x_n)\|_q$, and thus cannot simply consider $\left\| \tilde{R}(x_1, \ldots, x_n) \right\|_q$.

2. Second, even to lower bound $v(m, \epsilon, \delta, s)$, using $\left\| \tilde{R}(x_1, \ldots, x_n) \right\|_q$ as a upper bound for $\|R(x_1, \ldots, x_n)\|_q$ is not sufficiently strong. Below, we give a counter-example, i.e. a vector $x$, where $\left\| \tilde{R}(x_1, \ldots, x_n) \right\|_q$ is too large to recover a tight lower bound.

Thus, we cannot use the Hanson-Wright bound in this setting, and need to come up with a better bound on $\|R(x_1, \ldots, x_n)\|_q$ that does not implicitly replace Rademachers by gaussians. The second point is similar in flavor to the conceptual point made in [14], where a sign-consistent variant of sparse JL was analyzed using an *upper* bound for Rademacher quadratic forms. However, the bound in [14] also turns out to be loose in this setting and also can't be used to obtain either a sufficiently tight upper bound or a lower bound for $R(x_1, \ldots, x_n)$.

We now show point (2): that the Hanson-Wright bound is not sufficiently strong to obtain a lower bound $v(m, \epsilon, \delta, s)$. We consider $\tilde{R}(x_1, \ldots, x_n) = \frac{1}{s} \sum_{r=1}^{m} \sum_{i \neq j} \eta_{r,i} \eta_{r,j} g_{r,i} g_{r,j} x_i x_j$, as above, where the $g_{r,i}$ are i.i.d standard gaussians. We consider $p$ equal to $\ln(1/\delta)$ rounded up to the nearest even integer, and we consider a vector of the form $[v, \ldots, v, 0, \ldots, 0]$ where $\frac{1}{v^2}$ is an integer and $v \geq 0$. We show $\left\| \tilde{R}(v, \ldots, v, 0, \ldots, 0) \right\|_p \gtrsim \omega(\epsilon)$ for a certain $v$ value, where we know it to be true that $\|R(v, \ldots, v, 0, \ldots, 0)\|_p \lesssim \epsilon$.

Let's consider a vector $[v, \ldots, v, 0, \ldots, 0]$ where $\frac{1}{v^2}$ is an integer and $v \geq 0$. We apply the Hanson-Wright bound (which is tight for gaussians) to obtain:

$$\left\| \tilde{R}(v, \ldots, v, 0, \ldots, 0) \right\|_p \gtrsim p v^2 \left\| \sup_{\|x\|_2, \|y\|_2 \leq 1} \sum_{r=1}^{m} \sum_{1 \leq i \neq j \leq N} \eta_{r,i} \eta_{r,j} x_{r,i} y_{r,j} \right\|_p$$

$$\geq p v^2 \left\| \sup_{\|x\|_2, \|y\|_2 \leq 1} \sum_{1 \leq i \neq j \leq N} \eta_{1,i} \eta_{1,j} x_i y_j \right\|_p.$$

Let $M = \sum_{i=1}^{N} \eta_{1,i}$. Let $S \subseteq [N]$ be the set of indices where $\eta_{1,i} = 1$. We can set the vector to $x_i = y_i = \frac{1}{\sqrt{M}}$ for all $i \in S$ and 0 elsewhere. This gives us:

$$\left\| \sup_{\|x\|_2, \|y\|_2 \leq 1} \sum_{1 \leq i \neq j \leq N} \eta_{1,i} \eta_{1,j} x_i y_j \right\|_p \geq \|M - 1\|_p = \left\| \sum_{i=1}^{N} \eta_{1,i} - 1 \right\|_p \gtrsim \left\| I_{\sum_{i=1}^{N} \eta_{1,i} \geq 2} \sum_{i=1}^{N} \eta_{1,i} \right\|_p.$$

We can expand out this moment to obtain:

$$\mathbb{E}\left[\left(I_{\sum_{i=1}^{N}\eta_{1,i}\geq 2}\sum_{i=1}^{N}\eta_{1,i}\right)^{p}\right] \geq C^{p}\sum_{M=2}^{p}\binom{N}{M}M^{p}\left(\frac{s}{m}\right)^{M}\left(1-\frac{s}{m}\right)^{M}$$

$$\geq C^{p}\sum_{M=2}^{p}\left(\frac{N}{p}\right)^{M}\left(\frac{p}{M}\right)^{M}M^{p}\left(\frac{s}{m}\right)^{M}\left(1-\frac{s}{m}\right)^{M}$$

$$= C^{p}\sum_{M=2}^{p}\left(\frac{s}{pmv^{2}}\right)^{M}M^{p}\left(\frac{p}{M}\right)^{M}\left(1-\frac{s}{m}\right)^{M}.$$

Since $M \leq p$, we know that $\left(\frac{p}{M}\right)^{M} \geq 1$. Moreover, as long as $p \geq \frac{se}{mv^{2}}$, we know that $\left(1-\frac{s}{m}\right)^{M/p} \geq \left(1-\frac{s}{m}\right)^{N/p} \geq \left(1-\frac{s}{m}\right)^{\frac{m}{s}} \geq 0.3$. Thus we obtain a bound of

$$D^{p}\sum_{M=2}^{p}M^{p}\left(\frac{s}{pmv^{2}}\right)^{M}.$$

If $2 \leq \frac{p}{\ln(pmv^{2}/s)} \leq p$ (which can be written as $1 \leq \ln(pmv^{2}/s) \leq \frac{p}{2}$), then we know that:

$$pv^{2}\left\|\sup_{\|x\|_{2},\|y\|_{2}\leq 1}\sum_{1\leq i\neq j\leq N}\eta_{1,i}\eta_{1,j}x_{i}y_{j}\right\|_{p} \gtrsim \frac{p^{2}v^{2}}{\ln(pmv^{2}/s)}.$$

We show that when $s = 1$, the bound $v = \sqrt{\epsilon}\frac{\ln(\frac{m\epsilon}{p})}{p}$ will produce $\|R(v,\ldots,v,0,\ldots,0)\|_{p} \gtrsim \omega(\epsilon)$. At this $v$ value, we know that:

$$1 \leq \frac{pmv^{2}}{e} = \ln^{2}(\frac{m\epsilon}{p})\frac{m\epsilon}{pe}.$$

If we have that $\ln(\frac{m\epsilon}{p}) \leq \sqrt{p}$, then we know that $v \leq \frac{1}{\sqrt{p}}$ and $\ln(pmv^{2}) \leq \frac{p}{2}$. However, the bound

$$\frac{p^{2}v^{2}}{\ln(pmv^{2})} \gtrsim \frac{\epsilon\ln^{2}(m\epsilon/p)}{\ln(\frac{m\epsilon}{p})} \geq \epsilon\ln(\frac{m\epsilon}{p}) = \omega(\epsilon).$$

# C   Useful Moment Bounds

The key quadratic form bound for Rademachers that we use is:

**Lemma C.1** Let $T$ be an even integer, $\{\sigma_{i}\}_{1\leq i\leq n}$ be independent Rademachers, and $(Y_{i,j})_{1\leq i,j\leq n}$ be a $n \times n$ symmetric, nonnegative random matrix with zero diagonal (i.e. $Y_{i,i} = 0$) such that $\{Y_{i,j}\}_{1\leq i,j\leq n}$ is independent from $\{\sigma_{i}\}_{1\leq i\leq n}$. If $W_{i} = \sqrt{\sum_{1\leq j\leq n}Y_{i,j}^{2}}$, then:

$$\left\|\sum_{1\leq i,j\leq n}Y_{i,j}\sigma_{i}\sigma_{j}\right\|_{T} \simeq \left\|\sup_{\|b\|_{2},\|c\|_{2}\leq\sqrt{T},\|b\|_{\infty},\|c\|_{\infty}\leq 1}\sum_{1\leq i,j\leq n}Y_{i,j}b_{i}c_{j}\right\|_{T} + \left\|\sum_{1\leq i\leq T}W_{(i)} + \sqrt{T}\sqrt{\sum_{T<i\leq n}W_{(i)}^{2}}\right\|_{T}$$

where $W_{(1)} \geq W_{(2)} \geq \ldots \geq \ldots W_{(n)}$ is a permutation of $W_{1},\ldots,W_{n}$.

We derive Lemma C.1 from Latała's bound on Rademacher quadratic forms [21]. In fact, Latała shows moment bounds for much more general quadratic forms, but for the application to JL, we only need the following bound in the special case of Rademachers:

**Lemma C.2 ([21])** Let $T$ be an even natural number. Let $\sigma_{1},\ldots,\sigma_{n}$ be independent Rademachers and let $(a_{i,j})$ a symmetric matrix with zero diagonal. Then:

$$\left\|\sum_{1\leq i,j\leq n}a_{i,j}\sigma_{i}\sigma_{j}\right\|_{T} \simeq \left(\sup_{\|b\|_{2},\|c\|_{2}\leq\sqrt{T},\|b\|_{\infty},\|c\|_{\infty}\leq 1}\sum_{1\leq i,j\leq n}a_{i,j}b_{i}c_{j}\right) + \sum_{1\leq i\leq T}A_{(i)} + \sqrt{T}\sqrt{\sum_{T<i\leq n}(A_{(i)})^{2}}$$

where $A_i = \sqrt{\sum_{1 \leq j \leq n} a_{i,j}^2}$ and $A_{(1)} \geq A_{(2)} \ldots \geq \ldots A_{(n)}$ is a permutation of $A_1, \ldots, A_n$.

To prove Lemma $C.1$, we apply Lemma $C.2$ to the case where the $a_{i,j}$ are themselves random variables:

*Proof of Lemma C.1.* Let $Q = \sum_{1 \leq i \neq j \leq n} Y_{i,j} \sigma_i \sigma_j$. Applying Lemma $C.2$, we have that:

$$
\left( \mathbb{E}_{Y,\sigma}[Q^T] \right)^{1/T} = \left( \mathbb{E}_Y \mathbb{E}_\sigma [Q^T] \right)^{1/T}
$$

$$
= \left( \mathbb{E}_Y \left[ \left\| \sum_{1 \leq i \neq j \leq n} Y_{i,j} \sigma_i \sigma_j \right\|_T^T \right] \right)^{1/T}
$$

$$
\simeq \left\| \sup_{\|b\|_2, \|c\|_2 \leq \sqrt{T}, \|b\|_\infty, \|c\|_\infty \leq 1} \sum_{1 \leq i \neq j \leq n} Y_{i,j} b_i c_j + \sum_{1 \leq i \leq T} W_{(i)} + \sqrt{T} \sqrt{\sum_{T < i \leq n} W_{(i)}^2} \right\|_T
$$

$$
\simeq \left\| \sup_{\|b\|_2, \|c\|_2 \leq \sqrt{T}, \|b\|_\infty, \|c\|_\infty \leq 1} \sum_{1 \leq i \neq j \leq n} Y_{i,j} b_i c_j \right\|_T + \left\| \sum_{1 \leq i \leq T} W_{(i)} + \sqrt{T} \sqrt{\sum_{T < i \leq n} W_{(i)}^2} \right\|_T
$$

where the last line follows from the fact that the the $Y_{i,j}$ are nonnegative, so each term is nonnegative, so the triangle inequality results in at most a factor of 2 of gain. $\qquad \square$

Now, we consider linear forms of symmetric random variables. Theoretically, moments of these forms can be derived from Theorem 2 in [20] (a tight bound on moments of weighted sums of symmetric random variables). However, reducing the tight bound to the form that we want would require some simplifications. Instead, we give a direct proof of a weaker bound that is sufficiently tight for our setting.

**Proposition C.3** Suppose that $T \geq 1$ is an integer. Suppose that $Y_1, Y_2, \ldots, Y_n$ are i.i.d symmetric random variables and suppose that $x = [x_1, \ldots, x_n]$ satisfies $\|x\|_2 \leq 1$ and $\|x\|_\infty \leq v$. Then, we have that

$$
\left\| \sum_i Y_i x_i \right\|_{2T} \lesssim v \left( \sup_{1 \leq t \leq T} \frac{T}{t} \left( \frac{1}{Tv^2} \right)^{\frac{1}{2t}} \|Y_i\|_{2t} \right).
$$

*Proof of Proposition C.3.* Let $k = 2v \left( \sup_{1 \leq t \leq T} \frac{T}{t} \left( \frac{1}{Tv^2} \right)^{1/(2t)} \|Y_i\|_{2t} \right)$. Observe that

$$
\mathbb{E}\left[ \left( \frac{\sum_i Y_i x_i}{k^2} \right)^{2T} \right] = \sum_{d_1+d_2+\ldots+d_n=T, d_i \leq T} \frac{2T!}{2d_1! \ldots 2d_n!} \prod_{i=1}^n \mathbb{E}\left[ \left( \frac{Y_i x_i}{k} \right)^{2d_i} \right]
$$

$$
\leq C^T \sum_{d_1+d_2+\ldots+d_n=T, d_i \leq T} \frac{(2T)^{2T}}{(2d_1)^{2d_1} \ldots (2d_n)^{2d_n}} \prod_{i=1}^n \mathbb{E}\left[ \left( \frac{Y_i x_i}{k} \right)^{2d_i} \right]
$$

$$
\leq C^T \prod_{i=1}^n \sum_{0 \leq d_i \leq T} \frac{(2T)^{2d_i}}{(2d_i)^{2d_i}} \mathbb{E}\left[ \left( \frac{Y_i x_i}{k} \right)^{2d_i} \right]
$$

$$
= C^T \prod_{i=1}^n \left( 1 + \sum_{1 \leq d_i \leq T} \left( \frac{T x_i \|Y_i\|_{2d_i} v}{v d_i k} \right)^{2d_i} \right)
$$

Now, we use the fact that $|x_i| \leq v$ and the condition on $k$ to obtain that this is bounded by

$$
C^T \prod_{i=1}^n \left( 1 + \frac{x_i^2}{v^2} \sum_{1 \leq d_i \leq T} \left( \frac{T v \|Y_i\|_{2d_i}}{d_i k} \right)^{2d_i} \right) \leq C^T \prod_{i=1}^n \left( 1 + T x_i^2 \right) \leq C^T \prod_{i=1}^n e^{T x_i^2} \leq C^T e^T.
$$

$\qquad \square$

We now bound moments of squares of linear forms with a zero diagonal, i.e. $\sum_{i \neq j} Y_i Y_j x_i x_j$. This structure of random variable theoretically falls under the scope of Lemma C.1. However, as mentioned in the paper, the first term of C.1, which is an operator-norm-like term for an asymmetric random matrix in this setting, becomes intractable to manage. We give an alternate (weaker) upper bound that is both tractable to analyze and sufficiently tight for our setting. Our proof of this bound is similar to our proof of Proposition C.3 presented above. Since random variables with a zero diagonal are common in the JL literature [19, 3, 24], we believe this moment bound could be of broader use.

**Lemma C.4** Suppose that $Y_1, Y_2, \ldots, Y_n$ are i.i.d symmetric random variables and suppose that $x = [x_1, \ldots, x_n]$ satisfies $\|x\|_2 = 1$ and $\|x\|_\infty \leq v$. Let $T$ be an even natural number. Then, we have that

$$\left\| \sum_{i \neq j} Y_i Y_j x_i x_j \right\|_T \lesssim v^2 \left( \sup_{1 \leq t \leq T/2} \frac{T^2}{t^2} \left( \frac{1}{Tv^2} \right)^{1/t} \|Y_i\|_{2t}^2 \right).$$

*Proof of Lemma C.4.* Let $k = 2v \left( \sup_{1 \leq t \leq T/2} \frac{T}{t} \left( \frac{1}{Tv^2} \right)^{1/(2t)} \|Y_i\|_{2t} \right)$. Observe that

$$\mathbb{E} \left[ \left( \frac{\sum_{i \neq j} Y_i Y_j x_i x_j}{k^2} \right)^T \right] \leq \sum_{d_1 + d_2 + \ldots + d_n = T, d_i \leq T/2} \frac{2T!}{2d_1! \ldots 2d_n!} \prod_{i=1}^n \mathbb{E} \left[ \left( \frac{Y_i x_i}{k} \right)^{2d_i} \right]$$

$$\leq C^T \sum_{d_1 + d_2 + \ldots + d_n = T, d_i \leq T/2} \frac{(2T)^{2T}}{(2d_1)^{2d_1} \ldots (2d_n)^{2d_n}} \prod_{i=1}^n \mathbb{E} \left[ \left( \frac{Y_i x_i}{k} \right)^{2d_i} \right]$$

$$\leq C^T \prod_{i=1}^n \sum_{0 \leq d_i \leq T/2} \frac{(2T)^{2d_i}}{(2d_i)^{2d_i}} \mathbb{E} \left[ \left( \frac{Y_i x_i}{k} \right)^{2d_i} \right]$$

$$= C^T \prod_{i=1}^n \left( 1 + \sum_{1 \leq d_i \leq T/2} \left( \frac{T x_i \|Y_i\|_{2d_i} v}{v d_i k} \right)^{2d_i} \right)$$

Now, we use the fact that $|x_i| \leq v$ and the condition on $k$ to obtain that this is bounded by

$$C^T \prod_{i=1}^n \left( 1 + \frac{x_i^2}{v^2} \sum_{1 \leq d_i \leq T/2} \left( \frac{Tv \|Y_i\|_{2d_i}}{d_i k} \right)^{2d_i} \right) \leq C^T \prod_{i=1}^n \left( 1 + T x_i^2 \right) \leq C^T \prod_{i=1}^n e^{T x_i^2} \leq C^T e^T.$$

$\square$

Latała [20] gives the following nice bound on sums of i.i.d symmetric random variables that we use for combining bounds on rows $Z_r(x_1, \ldots, x_n)$ in Lemma 2.3.

**Lemma C.5 ([20])** Suppose that $q$ is an even natural number. Suppose that $Y_1, \ldots, Y_n$ are i.i.d symmetric random variables. Then:

$$\left\| \sum_{i=1}^n Y_i \right\|_q \lesssim \sup_{2 \leq T \leq q} \frac{q}{T} \left( \frac{n}{q} \right)^{1/T} \|Y_i\|_T.$$

We give a general result on moments of sums of certain (potentially correlated) sums of identically distributed random variables, that we use in proving Lemma 2.4.

**Proposition C.6** Let $Y_1, \ldots, Y_n$ be identically distributed (but not necessarily independent) random variables, such that the joint distribution is a symmetric function of $Y_1, \ldots, Y_n$ and for any integers $d_1, \ldots d_n \geq 0$, it is true that $\mathbb{E}[\prod_{1 \leq i \leq n} Y_i^{d_i}] \geq 0$. For any natural number $q$ and natural number $T$ that divides $q$, it is true that

$$\left\| \sum_{i=1}^n Y_i \right\|_q \geq T \left( \frac{n}{q} \right)^{T/q} \|Y_1 Y_2 \ldots Y_T\|_{q/T}^{1/T}$$

*Proof of Proposition C.6.* The proof follows from expanding $\mathbb{E}[(\sum_{i=1}^n Y_i)^q]$ and using the fact that $\mathbb{E}[\prod_{1 \le i \le n} Y_i^{d_i}] \ge 0$ so that we can restrict to a subset of the terms. By the symmetry of the joint distribution, we know that for $1 \le r_1 \ne r_2 \ne r_T \le n$, we know that $\mathbb{E}[Y_{r_1}^{q/T} \dots Y_{r_T}^{q/T}] = \mathbb{E}[Y_1^{q/T} \dots Y_T^{q/T}]$. The number of terms of the form $\mathbb{E}[Y_{r_1}^{q/T} \dots Y_{r_T}^{q/T}]$ in $\mathbb{E}[(\sum_{i=1}^n Y_i)^q]$ is:

$$\binom{n}{T}\binom{q}{q/T, q/T, \dots, q/T} \ge C^q \left(\frac{n}{T}\right)^T \frac{q!}{((q/T)!)^T} \ge C^q \left(\frac{n}{T}\right)^T T^q \ge C_2^q \left(\frac{n}{q}\right)^T \left(\frac{q}{T}\right)^T T^q \ge C'^q \left(\frac{n}{q}\right)^T T^q.$$

This implies that

$$\mathbb{E}\left[\left(\sum_{i=1}^n Y_i\right)^q\right] \ge C'^q \left(\frac{n}{q}\right)^T T^q \mathbb{E}\left[Y_1^{q/T} \dots Y_T^{q/T}\right]$$

and the statement follows from taking $1/q$th powers. $\qquad\square$

We prove a lemma involving the Paley-Zygmund inequality applied to $p$th moments, that we use implicitly in the proof of the upper bound in Theorem 1.5.

**Lemma C.7** Suppose that $K > 0$ and $Z$ is a nonnegative random variable, such that $\|Z\|_q \ge 2K$ and $\|Z\|_{2q}$ is finite. Then,

$$\mathbb{P}[Z > K] \ge 0.25 \left(\frac{\|Z\|_q}{\|Z\|_{2q}}\right)^{2q}.$$

We use the Paley-Zygmund inequality, which says the following:

**Lemma C.8 (Paley-Zygmund)** Suppose that $Z$ is a nonnegative random variable with finite variance. Then,

$$\mathbb{P}[Z > 2^{-1}\mathbb{E}[Z]] \ge \frac{\mathbb{E}[Z]^2}{4\mathbb{E}[Z^2]}.$$

*Proof of Lemma C.7.* We apply Lemma C.8 to $Z^p$ to obtain that:

$$\mathbb{P}[Z^p > 2^{-1}\mathbb{E}[Z^p]] \ge 0.25\frac{\mathbb{E}[Z^p]^2}{\mathbb{E}[Z^{2p}]} = 0.25\left(\frac{\|Z\|_p}{\|Z\|_{2p}}\right)^{2p}.$$

If $\|Z\|_p \ge 2K$, then we know that

$$\mathbb{P}[Z > K] = \mathbb{P}[Z^p > K^p] \ge \mathbb{P}[Z^p > 2^{-p}\mathbb{E}[Z^p]] \ge \mathbb{P}[Z^p > 2^{-1}\mathbb{E}[Z^p]]$$

and then we can apply the above result. $\qquad\square$

# D  Proofs of Lemma 2.1 and Lemma 2.2

We analyze the moments of $Z_r(x_1, \dots, x_n)$, proving Lemma 2.2 and Lemma 2.1. Our lower bound in Lemma 2.2 holds for $\|Z_r(v, \dots, v, 0, \dots, 0)\|_q$ as well as $\left\|Z_r(v, \dots, v, 0, \dots, 0)I_{\sum_{i=1}^{1/v^2} \eta_{r,i}=2}\right\|_T$ (for technical reasons discussed in Appendix E). Our upper bound in Lemma 2.1 holds for $\|Z_r(x_1, \dots, x_n)\|_q$. In Appendix D.1, we prove Lemma 2.2. In Appendix D.2, we prove Lemma 2.1.

## D.1  Proof of Lemma 2.2

The key ingredient of the proof is Lemma C.1 (for Rademacher quadratic forms). We can view $Z_r(v, \dots, v, 0, \dots, 0)$ as the following quadratic form:

$$Z_r(v, \dots, v, 0, \dots, 0) = v^2 \sum_{1 \le i \ne j \le N} \eta_{r,i}\eta_{r,j}\sigma_{r,i}\sigma_{r,j},$$

where $N = \frac{1}{v^2}$. Since the support of $\eta_{r,i}$ is $\{0, 1\}$ and due to symmetry of this random variable, it is tractable to analyze the expressions in Lemma C.1.

*Proof of Lemma 2.2.* First, we handle the case of $T = 2$:

$$\mathbb{E}[Z_r(v,\ldots,v,0,\ldots,0)]^2 = v^4 \mathbb{E}\left[\left(\sum_{i\neq j} \eta_{r,i}\eta_{r,j}\sigma_{r,i}\sigma_{r,j}\right)^2\right]$$

$$= 2v^4 \mathbb{E}\left[\sum_{i\neq j} \eta_{r,i}\eta_{r,j}\right] = 2v^4 \left(\frac{s}{m}\right)^2 N(N-1) \geq \frac{v^4 N^2 s^2}{m^2} = \frac{s^2}{m^2}$$

as desired.

Now we consider $T > 2$, and we prove a bound on $\|Z_1(v,\ldots,v,0,\ldots,0)\|_T$. We see that $\|Z_1(v,\ldots,v,0,\ldots,0)\|_T = v^2 \left\|\sum_{i\neq j} \eta_{1,i}\eta_{1,j}\sigma_{1,i}\sigma_{1,j}\right\|_T$. Fix $1 \leq M \leq \min(N,T)$. We use Lemma C.1 with $Y_{i,j} = \eta_{1,i}\eta_{1,j}I_{M=\sum_{k=1}^{N}\eta_{1,k}}$ to compute $\left\|\sum_{i\neq j} \eta_{1,i}\eta_{1,j}\sigma_{1,i}\sigma_{1,j}I_{M=\sum_{k=1}^{N}\eta_{1,k}}\right\|_T$. We will then aggregate over $2 \leq M \leq T$ and not even count $M = 1$ or $T < M \leq N$. We only use the operator-norm-like term in Lemma C.1. Observe that

$$I_{M=\sum_{k=1}^{N}\eta_{1,k}} \sup_{\|b\|_2,\|c\|_2\leq\sqrt{T},\|b\|_\infty,\|c\|_\infty\leq 1} \sum_{i\neq j} \eta_{1,i}\eta_{1,j}b_ic_j$$

is equal to

$$I_{M=\sum_{k=1}^{N}\eta_{1,k}} \sup_{\|b\|_2,\|c\|_2\leq\sqrt{T},\|b\|_\infty,\|c\|_\infty\leq 1} \sum_{i,j|\eta_{1,i}=1,\eta_{1,j}=1} b_ic_j \geq I_{M=\sum_{k=1}^{N}\eta_{1,k}} M(M-1),$$

where we set $b_i = 1$ on all $i$ such that $\eta_{1,i} = 1$ and $c_j = 1$ on all $j$ such that $\eta_{1,j} = 1$.

Since the events $M = \sum_{k=1}^{N}\eta_{1,k}$ are disjoint across different $M$ values, we know that:

$$\left\|\sum_{i\neq j}\eta_{1,i}\eta_{1,j}\sigma_{1,i}\sigma_{1,j}\right\|_T \gtrsim \left(\sum_{M=2}^{\min(T,N)}\left\|\sum_{i\neq j}\eta_{1,i}\eta_{1,j}\sigma_{1,i}\sigma_{1,j}I_{M=\sum_{k=1}^{N}\eta_{1,k}}\right\|_T^T\right)^{1/T}$$

$$\gtrsim \left(\sum_{M=2}^{\min(T,N)}\left\|I_{M=\sum_{k=1}^{N}\eta_{1,k}}\sup_{\|b\|_2,\|c\|_2\leq\sqrt{T},\|b\|_\infty,\|c\|_\infty\leq 1}\sum_{i\neq j}\eta_{1,i}\eta_{1,j}b_ic_j\right\|_T^T\right)^{1/T}$$

$$\gtrsim \left(\sum_{M=2}^{\min(T,N)}\left\|I_{M=\sum_{k=1}^{N}\eta_{1,k}}M^2\right\|_T^T\right)^{1/T}$$

$$= \left(\sum_{M=2}^{\min(T,N)}\mathbb{P}[M=\sum_{i=1}^{N}\eta_{1,i}]M^{2T}\right)^{1/T}$$

$$= \left(\sum_{M=2}^{\min(T,N)}\binom{N}{M}\left(\frac{s}{m}\right)^M\left(1-\frac{s}{m}\right)^{N-M}M^{2T}\right)^{1/T}$$

$$\gtrsim \left(\sum_{M=2}^{\min(T,N)}\left(\frac{Ns}{mT}\right)^M\left(\frac{T}{M}\right)^M\left(1-\frac{s}{m}\right)^{N-M}M^{2T}\right)^{1/T}$$

$$\gtrsim \left(\sum_{M=2}^{\min(T,N)}\left(\frac{s}{mTv^2}\right)^M\left(1-\frac{s}{m}\right)^{N-M}M^{2T}\right)^{1/T}$$

$$\gtrsim \left(\sum_{M=2}^{\min(T,N)}\left(\frac{s}{mTv^2}\right)^M M^{2T}\right)^{1/T},$$

where the last line follows from the fact that since $T \geq \frac{se}{mv^2}$ and $s \leq m/e$, we know that:

$$\left(1 - \frac{s}{m}\right)^{\frac{N-M}{T}} \geq \left(1 - \frac{s}{m}\right)^{\frac{N}{T}} \geq \left(1 - \frac{s}{m}\right)^{\frac{Nmv^2}{se}} \geq \left(1 - \frac{s}{m}\right)^{\frac{m}{s}} \geq 0.25.$$

Setting $t = T/M$, we obtain, up to constants:

$$\sup_{2 \leq M \leq \min(T,N)} \left(\frac{s}{mTv^2}\right)^{M/T} M^2 = \sup_{\max(1, T/N) \leq t \leq T/2} \left(\frac{T^2}{t^2}\right) \left(\frac{s}{mTv^2}\right)^{1/t}.$$

We can take a derivative to obtain the two expressions in the lemma statement at the following regimes of parameters: $\max(1, Tv^2) \leq \ln(Tmv^2/s) \leq T$ and $\ln(Tmv^2/s) > T$. The second regime aligns with the lemma statement. Thus it suffices to show that when $v \leq \frac{\sqrt{\ln(m/s)}}{\sqrt{T}}$, it is true that $Tv^2 \leq \ln(Tmv^2/s)$. This is a straightforward calculation[1].

Now, let's consider the case where we want to bound $\left\|Z_1(v, \ldots, v, 0, \ldots, 0)I_{\sum_{k=1}^{N} \eta_{1,k}=2}\right\|_T$. It follows from the above calculations, without taking the sum that we obtain a lower bound of

$$\left(\binom{N}{2}\left(\frac{s}{m}\right)^2\left(1 - \frac{s}{m}\right)^{N-2}\right)^{1/T} \gtrsim \left(\frac{s}{mTv^2}\right)^{2/T}.$$

$\square$

## D.2 Proof of Lemma 2.1

In the paper, we discussed the tractability issues with using the general quadratic form moment bound Lemma $C.1$ to upper bound $\|Z_r(x_1, \ldots, x_n)\|_q$. Thus, we require simpler bounds that are easier to analyze. Linear forms naturally arise in the upper bound since $Z_r(x_1, \ldots, x_n) = \left(\sum_{1 \leq i \leq n} \eta_{r,i}\sigma_{r,i}x_i\right)^2 - \sum_{1 \leq i \leq n} \eta_{r,i}x_i^2 \leq \left(\sum_{1 \leq i \leq n} \eta_{r,i}\sigma_{r,i}x_i\right)^2$. However, it turns out that a vanilla linear form bound (e.g. Proposition C.3) here is weak due to the loss arising from ignoring the $\sum_{1 \leq i \leq n} \eta_{r,i}x_i^2$ term. Thus, we use Lemma C.4 (our generalized bound tailored to squares of linear forms with a zero diagonal) to obtain:

**Lemma D.1** If $\|x\|_\infty \leq v$ and $\|x\|_2 \leq 1$, then we have that:

$$\|Z_r(x_1, \ldots, x_n)\|_T = \left\|\sum_{i \neq j} \eta_{r,i}\eta_{r,j}\sigma_{r,i}\sigma_{r,j}x_ix_j\right\|_T \lesssim v^2 \left(\sup_{1 \leq t \leq T/2} \frac{T^2}{t^2}\left(\frac{s}{mTv^2}\right)^{1/t}\right).$$

*Proof.* This can be seen by simply taking $Y_i = \eta_{r,i}\sigma_{r,i}$ in Lemma $C.4$. $\square$

It turns out that using only this bound would lose the $m \geq s \cdot e^{\Theta\left(\max(1, \frac{p\epsilon^{-1}}{s})\right)}$ branch in the lower bound on $v(m, \epsilon, \delta, s)$ in Theorem 1.5. The lower bound on moments of $\|Z_r(v, \ldots, v, 0, \ldots, 0)\|_T$ in Lemma 2.2 sheds light on where this loss may be arising. We see that the problematic case is when $v \geq \frac{\sqrt{\ln(m/s)}}{\sqrt{T}} =: v_1$, and so we require a new bound for this regime. Since the vector $[v_1, \ldots, v_1, 0, \ldots, 0]$ is in $S_v$ when $v_1 \leq v$, we can't hope to beat the bound of $\|Z_r(v_1, \ldots, v_1, 0, \ldots, 0)\|_T \gtrsim \frac{T^2v_1^2}{\ln^2(Tmv_1^2/s)} \simeq \frac{T}{\ln(m/s)}$ from Lemma 2.2. We show that we can match this value:

**Lemma D.2** Suppose that $x = [x_1, \ldots, x_n]$ satisfies $\|x\|_2 = 1$ and $\|x\|_\infty < v$. If $s \leq m/e$, $T \geq \frac{se}{mv^2}$, $T \geq 3$, $T \geq \ln(mv^2/s)$, then:

$$\|Z_r(x_1, \ldots, x_n)\|_T = \left\|\sum_{i \neq j} \eta_{r,i}\eta_{r,j}\sigma_{r,i}\sigma_{r,j}x_ix_j\right\|_T \leq \left\|\sum_i \eta_{r,i}\sigma_{r,i}x_i\right\|_{2T}^2 \lesssim \frac{T}{\ln(m/s)}.$$

The proof of this bound requires a new technique that handles larger $|x_i|$ entries, while still managing the many smaller $|x_i|$ that are still allowed to be present. We separate out $|x_i| \geq \frac{\sqrt{\ln(m/s)}}{\sqrt{T}}$ and $|x_i| \leq \frac{\sqrt{\ln(m/s)}}{\sqrt{T}}$. In the quadratic form formulation of $Z_r(x_1, \ldots, x_n)$, this separation cannot be carried out, since there would be cross-terms between $|x_i| \geq \frac{\sqrt{\ln(m/s)}}{\sqrt{T}}$ and $|x_i| \leq \frac{\sqrt{\ln(m/s)}}{\sqrt{T}}$. As a result, we require the linear form bound (Proposition $C$.3) for $|x_i| \leq \frac{\sqrt{\ln(m/s)}}{\sqrt{T}}$, and it turns out to be sufficiently tight in this regime.

*Proof of Lemma D.2.* WLOG, assume that $|x_1| \geq |x_2| \geq \ldots \geq |x_n|$. Let $P = \left\lceil \frac{T}{\ln(m/s)} \right\rceil$. We know that

$$\left\| \sum_i \eta_{r,i} \sigma_{r,i} x_i \right\|_{2T} \leq \left\| \sum_{1 \leq i \leq P} \eta_{r,i} \sigma_{r,i} x_i \right\|_{2T} + \left\| \sum_{i > P} \eta_{r,i} \sigma_{r,i} x_i \right\|_{2T}.$$

For $1 \leq i \leq P$, we use the bound $|\sum_{i=1}^p \eta_{r,i} \sigma_{r,i} x_i| \leq \sum_{i=1}^p |x_i| \leq \sqrt{\left\lceil \frac{T}{\ln(m/s)} \right\rceil} \leq 2\sqrt{\frac{T}{\ln(m/s)}}$. For the remaining terms, we take $Y_i = \eta_{r,i} \sigma_{r,i}$ in Proposition $C$.3 to obtain the following upper bound[2] for $|x_i| \leq v' := \frac{\sqrt{\ln(m/s)}}{\sqrt{T}}$ and $\|x\|_2 \leq 1$:

$$\left\| \sum_i \eta_{r,i} \sigma_{r,i} x_i \right\|_{2T} \lesssim v' \left( \sup_{1 \leq t \leq T} \frac{T}{t} \left( \frac{s}{mTv'^2} \right)^{\frac{1}{2t}} \right).$$

Based on the conditions in this lemma statement, we know that $\frac{mTv'^2}{s} = \frac{mT\ln(m/s)}{sT} = \frac{m}{s\ln(m/s)} \geq e$. Thus taking a derivative, we obtain that this can be upper bounded by taking $t = \ln(mTv'^2/s)$ which yields:

$$\frac{Tv'}{\ln(mTv'^2/s)} = \frac{Tv'}{\ln(m/s) - \ln\ln(m/s)} \leq \frac{Tv'}{0.5\ln(m/s)} = 2\frac{\sqrt{T}}{\sqrt{\ln(m/s)}}.$$

$\square$

Finally, combining Lemma $D$.1 and Lemma $D$.2 yields Lemma 2.1:

*Proof of Lemma 2.1.* We apply Lemma $D$.1 at $T = 2$ to directly obtain $\frac{Ts}{m}$, and for $T \geq 3$, we apply Lemma $D$.1 and take a derivative to obtain:

$$\left\| \sum_{i \neq j} \eta_{r,i} \eta_{r,j} \sigma_{r,i} \sigma_{r,j} x_i x_j \right\|_T \lesssim v^2 \begin{cases} \frac{Ts}{mv^2}, & \text{for } se \geq mTv^2 \\ \frac{T^2}{\ln(mTv^2/s)^2}, & \text{for } se \leq mTv^2, \ln(Tmv^2/s) \leq T \\ \left( \frac{s}{mTv^2} \right)^{2/T}, & \text{for } \ln(Tmv^2/s) \geq T, se \leq mTv^2 \end{cases}.$$

To obtain the desired bound, we also include the bound from Lemma $D$.2 in the middle regime. $\square$

# E  Combining rows to bound $\|R(x_1, \ldots, x_n)\|_q$

Now, we show to move from bounds on moments of individual rows (i.e. $Z_r(x_1, \ldots, x_n)$) to bounds on moments of $R(x_1, \ldots, x_n)$. In Appendix E.1, we obtain an upper bound on $\|R(x_1, \ldots, x_n)\|_q$, thus proving Lemma 2.3. In Appendix E.2, we obtain a lower bound on $\|R(x_1, \ldots, x_n)\|_q$, thus proving Lemma 2.4.

## E.1  Proof of Lemma 2.3

Since the $\eta_{r,i}$ are negatively correlated, we can always upper bound the moments of $R(x_1,\ldots,x_n)$ by the case of a sum of *independent* random variables when $q$ is even[3] $Z_r'(x_1,\ldots,x_n) \sim Z_r(x_1,\ldots,x_n)$.

$$s \cdot \|R(x_1,\ldots,x_n)\|_q \le \left\|\sum_{r=1}^m Z_r(x_1,\ldots,x_n)\right\|_q \le \left\|\sum_{r=1}^m Z_r'(x_1,\ldots,x_n)\right\|_q \lesssim \sup_{2\le T\le q} \frac{q}{T}\left(\frac{m}{q}\right)^{1/T} \|Z_1(x_1,\ldots,x_n)\|_T,$$
$$(2)$$

where the last inequality follows from Lemma $C$.5. Thus, it remains to analyze the sup expression. It turns out that each regime of bounds in Lemma 2.1 collapses to one value, so the different regimes in Lemma 2.1 correspond to different parts of the max expressions in Lemma 2.3. Depending on the parameters, some of these regimes may not exist, as is reflected by branches of the max expression sometimes vanishing in Lemma 2.1. We defer the computation to Appendix F.

## E.2  Proof of Lemma 2.4

Moving from a lower bound on the moments of individual rows given by Lemma 2.2 to moments of $R(v,\ldots,v,0,\ldots,0)$ is more delicate. Unlike in the upper bound, the negative correlations between random variables require some care to handle, even with the simplification that the $s$ nonzero entries in a column are chosen uniformly at random. For example, the conditional distribution of $\eta_{s+1,1} \mid \eta_{1,1} = \eta_{2,1} = \ldots = \eta_{s,1} = 1$ is 0, while the marginal distribution of $\eta_{s+1,1}$ has expectation $s/m$. One aspect that simplifies our analysis is that we *know* from our proof of Lemma 2.3 which moments of $Z_r(x_1,\ldots,x_n)$ are critical in the sup expression in (2). We only need to account for these particular moments in our lower bound approach. It turns out that the three critical values are $q/T = 2$, $q/T = q$, and $q/T = \ln(qmv^4/s^2)$.

For $q/T = q$, where rows are isolated, we can directly obtain a bound from Lemma $C$.6 and Lemma 2.2 to obtain.

**Lemma E.1** Suppose $\mathcal{A}_{s,m,n}$ is a uniform sparse JL distribution. Suppose that $q$ is even, $s \le m/e$, $q \ge \frac{se}{mv^2}$, $1 \le \ln(qmv^2/s) \le q$, $v \le \frac{\sqrt{\ln(m/s)}}{\sqrt{q}}$ and $\frac{1}{v^2}$ is an even integer. Then it is true that:

$$\|R(v,\ldots,v,0,\ldots,0)\|_q \gtrsim \frac{q^2 v^2}{s\ln^2\left(\frac{qmv^2}{s}\right)}.$$

*Proof.* By Lemma $C$.6 with $T = 1$, we have that:

$$\|R(v,\ldots,v,0,\ldots,0)\|_q \ge \frac{m^{1/q}}{s}\|Z_1(v,\ldots,v,0,\ldots,0)\|_q \ge \frac{1}{s}\|Z_1(v,\ldots,v,0,\ldots,0)\|_q.$$

Now, we apply Lemma 2.2 to obtain the desired expression.  □

For $q/T = 2$ and $q/T = \ln(qmv^4/s^2)$, we make use of the Lemma E.2 that relates moments of products of rows to products of moments of rows by taking advantage of either $s$ and $\frac{1}{v^2}$ being sufficiently large. The method essentially uses a counting argument to show that not too many terms vanish as a result of negative correlations, and requires adding in an indicator for the number of nonzero entries in a row being 2 for some cases (which is sufficient to prove Lemma 2.4).

**Lemma E.2** Suppose $\mathcal{A}_{s,m,n}$ is a uniform sparse JL distribution. If $1 \le T \le q/2$ is an integer, $q/T$ is an even integer, $\frac{1}{v^2}$ is an even integer, and $2Tv^2 \le s$, then:

$$\left\|\prod_{i=1}^T Z_i(v,\ldots,v,0,\ldots,0)\right\|_{q/T}^{1/T} \gtrsim \begin{cases} \|Z_1(v,\ldots,v,0,\ldots,0)\|_2 & \text{if } T = q/2 \\ \left\|Z_1(v,\ldots,v,0,\ldots,0)I_{\sum_{i=1}^N \eta_{1,i}=2}\right\|_{q/T} & \text{if } 1 \le T \le q/2 \end{cases}.$$

We defer the proof to Appendix G.

Now we can use Lemma $C$.6 coupled with Lemma $E$.2 and Lemma 2.2 to handle the cases of $q/T = 2, \ln(qmv^4/s^2)$ and obtain the following bounds. For $q/T = 2$, we obtain:

**Lemma E.3** Suppose $\mathcal{A}_{s,m,n}$ is a uniform sparse JL distribution. If $q$ is an even integer, $\frac{qv^2}{s} \leq 1$, and $\frac{1}{v^2}$ is an even integer, then it is true that:

$$\|R(v,\ldots,v,0,\ldots,0)\|_q \gtrsim \left(\frac{q}{m}\right)^{1/2}.$$

*Proof of Lemma E.3.* Take $T = \frac{q}{2}$ and $qv^2 \leq s$. By Lemma E.2 and Lemma C.6, we have that:

$$\|R(v,\ldots,v,0,\ldots,0)\|_q \gtrsim \frac{q}{s}\left(\frac{m}{q}\right)^{1/2}\|Z_1(v,\ldots,v,0,\ldots,0)\|_2.$$

Now, by Lemma 2.2, we can see that $\|Z_1(v,\ldots,v,0,\ldots,0)\|_2 \gtrsim \frac{s}{m}$. Thus, our bound becomes:

$$\frac{q}{s}\left(\frac{m}{q}\right)^{1/2}\frac{s}{m} = \left(\frac{q}{m}\right)^{1/2}.$$

$\square$

For $q/T = \ln(qmv^4/s^2)$, we similarly obtain the following bound using Lemma $C.6$ coupled with Lemma $E.2$.

**Lemma E.4** Suppose $\mathcal{A}_{s,m,n}$ is a uniform sparse JL distribution. Suppose that $q$ is a power of 2, $s \leq m/e$, $2qv^2 \leq 0.5s\ln(qmv^4/s^2)$, $\frac{1}{v^2}$ is even, $2 \leq \ln(qmv^4/s^2) \leq q$, and $m \geq q$. Then it is true that:

$$\|R(v,\ldots,v,0,\ldots,0)\|_q \gtrsim \frac{qv^2}{s\ln(\frac{qmv^4}{s^2})}.$$

*Proof.* Let's let $f(x)$ be the function that rounds $x$ to the nearest power of 2. By the conditions, we know that $2 \leq f(\ln(qmv^4/s^2)) \leq q$. Now, we want the condition $2qv^2 \leq sf(\ln(qmv^4/s^2))$ to be satisfied. If $f(\ln(qmv^4/s^2)) \geq \ln(qmv^4/s^2)$, then this is implied by $2qv^2 \leq s\ln(qmv^4/s^2) = s\max(\ln(qmv^4/s^2),2)$, which is a strictly weaker condition than the one given in the lemma statement. If $f(\ln(qmv^4/s^2)) \leq \ln(qmv^4/s^2)$, then $f(\ln(qmv^4/s^2)) \geq 0.5\ln(qmv^4/s^2)$ and so $2qv^2 \leq 0.5s\ln(qmv^4/s^2) \leq sf(\ln(qmv^4/s^2))$ gives the desired condition.

We use the fact that $\ln(qmv^4/s^2)/2 \leq f(\ln(qmv^4/s^2)) \leq 2\ln(qmv^4/s^2)$. We apply Lemma E.2 and Lemma C.6, with $T = \frac{q}{f(\ln(qmv^4/s^2))}$ and Lemma 2.2 to see that if we have the additional condition that $f(\ln(qmv^4/s^2)) \geq \frac{se}{mv^2}$, then we know that:

$$\|R(v,\ldots,v,0,\ldots,0)\|_q \gtrsim \frac{q}{sf(\ln(\frac{qmv^4}{s^2}))}\left(\frac{m}{q}\right)^{1/f(\ln(\frac{qmv^4}{s^2}))}\|Z_1(v,\ldots,v,0,\ldots,0)I_{M=2}\|_{f(\ln(\frac{qmv^4}{s^2}))}$$

$$\gtrsim \frac{qv^2}{2s\ln(\frac{qmv^4}{s^2})}\left(\frac{m}{q}\right)^{1/f(\ln(\frac{qmv^4}{s^2}))}\left(\frac{s}{mf(\ln(\frac{qmv^4}{s^2}))v^2}\right)^{2/f(\ln(\frac{qmv^4}{s^2}))}$$

$$= \frac{qv^2}{2s\ln(\frac{qmv^4}{s^2})}\left(\frac{s^2}{qmv^4}\right)^{1/f(\ln(\frac{qmv^4}{s^2}))}\left(\frac{1}{f(\ln(qmv^4/s^2))^2}\right)^{1/f(\ln(qmv^4/s^2))}$$

$$\gtrsim \frac{qv^2}{s\ln(\frac{qmv^4}{s^2})}\left(\frac{s^2}{qmv^4}\right)^{\frac{2}{\ln(\frac{qmv^4}{s^2})}}.$$

$$\gtrsim \frac{qv^2}{s\ln(\frac{qmv^4}{s^2})}.$$

Now, we see that

$$\frac{mv^2}{se} = \sqrt{\frac{qmv^4}{s^2}}\frac{1}{e}\frac{\sqrt{m}}{\sqrt{q}} \geq \frac{\sqrt{m}}{\sqrt{q}} \geq 1.$$

This implies that $\frac{se}{mv^2} \leq 1$, so the condition of $f(\ln(qmv^4/s^2)) \geq 2 \geq \frac{se}{mv^2}$ is automatically satisfied. $\square$

With these bounds, Lemma 2.4 follows.

*Proof of Lemma* 2.4. We combine Lemma $E.1$, Lemma $E.3$, and Lemma $E.4$. $\square$

# F  Proofs of Auxiliary Lemmas for Lemma 2.3

First, we use Lemma $C.5$ and Lemma 2.1 to prove a upper bound $\|R(x_1, \ldots, x_q)\|_q$ that is not quite in the desired form for Lemma 2.3.

**Lemma F.1** Let $2 \leq q \leq m$ be an even integer and $|x_i| \leq v$ and $\|x\|_2 = 1$. If $\frac{se}{mv^2} \geq q$, then:

$$\|R(x_1, \ldots, x_n)\|_q \lesssim \alpha_1(q, v, s, m).$$

If $\ln(qmv^2/s) > q$ then we have

$$\|R(x_1, \ldots, x_n)\|_q \lesssim \max(\alpha_1(q, v, s, m), \alpha_2(q, v, s, m)).$$

In all other cases, we have that

$$\|R(x_1, \ldots, x_n)\|_q \lesssim \max(\alpha_1(q, v, s, m), \alpha_2(q, v, s, m), \min(\alpha_3(q, v, s, m), \alpha_4(q, v, s, m))).$$

The functions are defined as follows.

$$\alpha_1(q, v, s, m) = \frac{\sqrt{q}}{\sqrt{m}}$$

$$\alpha_2(q, v, s, m) = \begin{cases} \frac{eqv^2}{s \ln(qmv^4/s^2)} & \text{for } \ln(qmv^4/s^2) \geq 2 \\ \frac{\sqrt{q}}{\sqrt{m}} & \text{for } \ln(qmv^4/s^2) \leq 2 \end{cases}$$

$$\alpha_3(q, v, s, m) = \frac{qv^2e}{s} \sup_{T \leq q, T \geq \max(\frac{se}{mv^2}, 3, \ln(mv^2 T/s))} \frac{T}{\ln^2(mv^2 T/s)} \left(\frac{s}{qv^2}\right)^{1/T}$$

$$\alpha_4(q, v, s, m) = \frac{qe^2}{s \ln(m/s)} \begin{cases} 1 & \text{for } \ln(qmv^4/s^2) \geq 2 \\ \left(\frac{s}{qv^2}\right)^{1/\ln(mv^2/s)} & \text{else} \end{cases}$$

*Proof of Lemma F.1.* As we discussed in Appendix E, it suffices to bound

$$\frac{1}{s} \sup_{2 \leq T \leq q} \frac{q}{T} \left(\frac{m}{q}\right)^{1/t} \|Z_1(x_1, \ldots, x_n)\|_t.$$

Our bounds on $\|Z_1(x_1, \ldots, x_n)\|_t$ are based on Lemma 2.1. We split into cases based on the $T$ value, and how it separates into different cases in Lemma 2.1. Let

$$\beta_1(q, v, s, m) = \frac{1}{s} \sup_{T=2, 3 \leq T \leq \frac{se}{mv^2}} \frac{q}{T} \left(\frac{m}{q}\right)^{1/t} \|Z_1(x_1, \ldots, x_n)\|_t.$$

$$\beta_2(q, v, s, m) = \frac{1}{s} \sup_{\max(3, \frac{se}{mv^2}) \leq T \leq \ln(mv^2 T/s)} \frac{q}{T} \left(\frac{m}{q}\right)^{1/t} \|Z_1(x_1, \ldots, x_n)\|_t.$$

$$\beta_{34}(q, v, s, m) = \frac{1}{s} \sup_{T \geq \max(3, \frac{se}{mv^2}, \ln(mv^2 T/s))} \frac{q}{T} \left(\frac{m}{q}\right)^{1/t} \|Z_1(x_1, \ldots, x_n)\|_t.$$

Let $\beta_3$ branch arise when we use the $\frac{T^2 v^2}{\ln(T m v^2/s)^2}$ for the $\|Z_1(x_1, \ldots, x_n)\|_t$ bound, and let the $\beta_4$ branch arise when we use $\frac{T v^2}{s \ln(m/s)}$ for the $\|Z_1(x_1, \ldots, x_n)\|_t$ bound. Thus, we know that

$$\beta_{34}(q, v, s, m) \leq \min(\beta_3(q, v, s, m), \beta_4(q, v, s, m)).$$

Let's first consider $\frac{se}{mv^2} \geq q$. In this case, only the $\beta_1$ branch arises. Now, suppose that $\frac{se}{mv^2} < q$.

Suppose that $\ln(qmv^2/s) > q$. Then we show that the $\beta_{34}$ branch does not arise. It suffices to show that $\ln(T m v^2/s) > T$ for all $T \geq \frac{se}{mv^2}$. Let $x = T m v^2/s$. It suffices to show that $\frac{s}{mv^2} \frac{x}{\ln x} < 1$ for all $e \leq x \leq \frac{qmv^2}{s}$. Since $\frac{s}{mv^2} \frac{x}{\ln x} < 1$ at $x = \frac{qmv^2}{s}$ and this is an increasing function of $x$, we know that the condition is true.

We now produce bounds $\alpha_1(q, v, s, m), \ldots, \alpha_4(q, v, s, m)$ such that $\beta_i(q, v, sm) \lesssim \alpha_i(q, v, s, m)$, which is what we do for the remainder of the analysis.

First, we handle the $\beta_1(q, v, s, m)$ term. We see that

$$\beta_1(q, v, s, m) = \frac{1}{s} \sup_{2 \leq T \leq \frac{s}{mv^2}} \frac{q}{T} \left(\frac{m}{q}\right)^{1/T} \frac{Ts}{m} = \frac{1}{s} \frac{qs}{m} \left(\frac{m}{q}\right)^{1/T} \leq \frac{q}{m} \left(\frac{m}{q}\right)^{1/2} = \frac{\sqrt{q}}{\sqrt{m}}.$$

Now, we handle the $\beta_2(q, v, s, m)$ term. We obtain a bound for $\|Z_r\|_T \lesssim v^2 \left(\frac{s}{mTv^2}\right)^{2/T}$. The expression becomes:

$$\beta_2(q, v, s, m) = \frac{1}{s} \sup_{T \geq \max(\frac{se}{mv^2}, 3), T \leq \ln(mv^2 T/s)} \frac{qv^2}{T} \left(\frac{m}{q}\right)^{1/T} \left(\frac{s}{mTv^2}\right)^{2/T}$$

$$= \frac{1}{s} \sup_{T \geq \max(\frac{se}{mv^2}, 3), T \leq \ln(mv^2 T/s)} \frac{qv^2}{T} \left(\frac{s}{\sqrt{qm}Tv^2}\right)^{2/T}$$

$$\leq \frac{1}{s} \sup_{T \geq \max(\frac{se}{mv^2}, 3), T \leq \ln(mv^2 T/s)} \frac{qv^2}{T} \left(\frac{s^2}{qmv^4}\right)^{1/T}.$$

Suppose that $\ln(qmv^4/s^2) \geq 2$. In this case, we have that this expression is upper bounded by $T = \ln(qmv^4/s^2)$. When we plug this into the expression, we obtain $\frac{qv^2}{s \ln(qmv^4/s^2)}$. Otherwise, if $\ln(qmv^4/s^2) \leq 2$, then this expression is upper bounded by $T = 3$:

$$\frac{qv^2}{s} \left(\frac{s^2}{qmv^4}\right)^{1/3} = \frac{C_1 C_5 q^{2/3} v^{2/3}}{s^{1/3} m^{1/3}}.$$

We know that that $\frac{q^{2/3} v^{2/3}}{s^{1/3} m^{1/3}} \leq \frac{\sqrt{q}}{\sqrt{m}}$ because this reduces to

$$\frac{q^{1/6} v^{2/3} m^{1/6}}{s^{1/3}} = \left(\frac{qmv^4}{s^2}\right)^{1/6} \leq e^{1/3}.$$

Now, we handle the $\beta_4(q, v, s, m)$ term when $\ln(qmv^2/s) \leq q$.

$$\beta_4(q, v, sm) = \frac{1}{s} \sup_{T \geq \max(\frac{se}{mv^2}, 3, \ln(mv^2 T/s))} \frac{q}{T} \left(\frac{m}{q}\right)^{1/T} \frac{T}{\ln(m/s)}$$

$$\leq \sup_{T \geq \max(\frac{se}{mv^2}, 3, \ln(mv^2 T/s))} \frac{q}{s \ln(m/s)} \left(\frac{mv^2}{s}\right)^{1/T} \left(\frac{s}{qv^2}\right)^{1/T}$$

$$\leq \frac{qe}{s \ln(m/s)} \sup_{T \geq \max(\frac{se}{mv^2}, 3, \ln(mv^2 T/s))} \left(\frac{s}{qv^2}\right)^{1/T}$$

If $s \leq qv^2$, this is bounded by 1, and if $s \geq qv^2$, this is bounded by $\left(\frac{s}{qv^2}\right)^{1/\ln(mv^2/s)}$. We see that $\frac{s}{qv^2} \leq \frac{mv^2}{s}$, so $\left(\frac{s}{qv^2}\right)^{1/\ln(mv^2/s)} \leq \left(\frac{mv^2}{s}\right)^{1/\ln(mv^2/s)} \leq e$. Thus this is bounded by $\frac{qe^2}{s \ln(m/s)}$.

Now, we handle the $\beta_3(q, v, s, m)$ term. In this case, the expression becomes:

$$\beta_3(q, v, s, m) = \frac{1}{s} \sup_{T \geq \max(\frac{se}{mv^2}, 3, \ln(mv^2 T/s))} \frac{qv^2}{T} \left(\frac{m}{q}\right)^{1/T} \frac{T^2}{\ln^2(mv^2 T/s)}$$

$$\leq \sup_{T \geq \max(\frac{se}{mv^2}, 3, \ln(mv^2 T/s))} \frac{qv^2 T}{s \ln^2(mv^2 T/s)} \left(\frac{mv^2}{s}\right)^{1/T} \left(\frac{s}{qv^2}\right)^{1/T}$$

$$\leq \frac{qv^2e}{s} \sup_{T \geq \max(\frac{se}{mv^2}, 3, \ln(mv^2T/s))} \frac{T}{\ln^2(mv^2T/s)} \left(\frac{s}{qv^2}\right)^{1/T}$$

$\square$

We use some function bounding arguments to come with a simpler bound for $\alpha_3$ for sufficiently large $v$.

**Lemma F.2** Assume that $C_2 q^3 mv^4 \geq s^2$ for some $C_2 \geq 1$. Then it is true that

$$\frac{qv^2e}{s} \sup_{T \leq q, T \geq \frac{se}{mv^2}, 3, \ln(mv^2T/s)} \frac{T}{\ln^2(mv^2T/s)} \left(\frac{s}{qv^2}\right)^{1/T} \leq \frac{C_2^{1/3} q^2 v^2 e^5}{s \ln^2(mv^2q/s)}.$$

*Proof of Lemma F.2.* With the assumptions that we made we know that $\frac{s}{q^3 v^2 C_2^2} \leq \frac{mv^2}{s}$. This implies that our expression becomes:

$$E = \frac{qv^2e}{s} \sup_{T \leq q, T \geq \max(\frac{se}{mv^2}, 3, \ln(mv^2T/s))} \frac{T}{\ln^2(mv^2T/s)} \left(\frac{s}{qv^2}\right)^{1/T}$$

$$= \frac{qv^2e}{s} \sup_{T \leq q, T \geq \max(\frac{se}{mv^2}, 3, \ln(mv^2T/s))} C_2^{1/T} \frac{T}{\ln^2(mv^2T/s)} \left(\frac{s}{C_2 q^3 v^2}\right)^{1/T} q^{2/T}$$

$$\leq \frac{qv^2e^2}{s} C_2^{1/3} \sup_{T \leq q, T \geq \max(\frac{se}{mv^2}, 3, \ln(mv^2T/s))} \frac{T}{\ln^2(mv^2T/s)} q^{2/T}$$

.

It suffices to show that $\sup_{T \leq q, T \geq \max(\frac{se}{mv^2}, 3, \ln(mv^2T/s))} \frac{T}{\ln^2(mv^2T/s)} q^{2/T} \leq \frac{qe^3}{\ln^2(mv^2q/s)}$.

Let $T_{min}$ be the minimum $T$ such that $T \geq \max(\frac{se}{mv^2}, 3, \ln(mv^2T/s))$. We just need to bound

$$\sup_{T_{min} \leq T \leq q} \frac{Tq^{2/T}}{\ln^2(mv^2T/s)} \leq \max\left(\sup_{T_{min} \leq T \leq \ln q} \frac{Tq^{2/T}}{\ln^2(mv^2T/s)}, \sup_{\max(T_{min}, \ln q) \leq T \leq q} \frac{Tq^{2/T}}{\ln^2(mv^2T/s)}\right)$$

$$\leq \max\left(\sup_{T_{min} \leq T \leq \ln q} \frac{Tq^{2/T}}{\ln^2(mv^2T/s)}, e^2 \sup_{\max(T_{min}, \ln q) \leq T \leq q} \frac{T}{\ln^2(mv^2T/s)}\right)$$

First, we handle the second term. Let $Q = \frac{mv^2T}{s}$. We use that $T_{min} \geq \frac{se}{mv^2}$, so $\frac{mv^2T_{min}}{s} \geq e$ to conclude $Q \geq e$. We see that

$$e^2 \sup_{\max(T_{min}, \ln q) \leq T \leq q} \frac{T}{\ln^2(mv^2T/s)} \leq e^2 \frac{s}{mv^2} \sup_{e \leq Q \leq \frac{qmv^2}{s}} \frac{Q}{\ln^2(Q)}.$$

We see that setting $Q$ to its maximum value achieves within a factor of $e$ of the maximum value of $\frac{Q}{\ln^2(Q)}$. Thus, we obtain that this is upper bounded by $e^3 \frac{q}{\ln^2(mv^2q/s)}$.

Now, we just need to handle the first term. If $T_{min} \geq \ln q$, then this term doesn't exist. Let's take a log of the expression to obtain:

$$\ln\left(\frac{T}{\ln^2(mv^2T/s)}\right) = \ln T - 2 \ln \ln(mv^2T/s) + \frac{2}{T} \ln(q)).$$

The derivative is:

$$\frac{1}{T} - \frac{2}{T \ln(mv^2T/s)} - \frac{2}{T^2} \ln(q).$$

The sign of the derivative is the same as:

$$1 - \frac{2}{\ln(mv^2T/s)} - \frac{2 \ln q}{T}.$$

Since $T_{min} \geq \frac{se}{mv^2}$, we know that $\ln(mv^2T/s) \geq 0$. Thus, we know that $1 - \frac{2}{\ln(mv^2T/s)} \leq 1$. Since $T \leq \ln q$, we know that $\frac{\ln q}{T} \geq 1$, so $-\frac{2\ln q}{T} \leq -2$. Thus, the derivative is negative, so the sup is attained at $T_{min} = T$, where the expression is:

$$e^3 \frac{T_{min}q^{2/T_{min}}}{\ln^2(mv^2T_{min}/s)} \leq e^3 \frac{(\ln q)q^{2/3}}{\ln^2(mv^2T_{min}/s)} \leq e^3 \frac{q^{3/4}}{\ln^2(mv^2T_{min}/s)}.$$

Thus, to upper bound by $\frac{q}{\ln^2(mv^2q/s)}$, it suffices to show:

$$\frac{\ln^2(mv^2q/s)}{\ln^2(mv^2T_{min}/s)} \leq q^{0.25}.$$

If $\frac{s}{mv^2} \leq 1$, the ratio is at most

$$\frac{\ln(mv^2q/s)}{\ln(mv^2T_{min}/s)} \leq \frac{\ln(mv^2/s) + \ln q}{\ln(mv^2/s) + \ln T_{min}} \leq \frac{\ln q}{\ln e} = \ln q \leq q^{0.25}.$$

If $\frac{s}{mv^2} \geq 1$, then $qmv^2/s \leq q$. Using this and $\frac{mv^2T_{min}}{s} \geq e$, we know:

$$\frac{\ln(mv^2q/s)}{\ln(mv^2T_{min}/s)} \leq \frac{\ln(q)}{\ln(e)} = \ln q \leq q^{0.25}.$$

$\square$

Now, we combine Lemma $F.1$ and Lemma $F.2$ to prove Lemma 2.3.

*Proof of Lemma* 2.3. First, we compute the second moment by hand:

$$\mathbb{E}[R(x_1,\ldots,x_n)]^2 = \frac{1}{s^2}\mathbb{E}\left[\left(\sum_{r=1}^{m}\sum_{i\neq j}\eta_{r,i}\eta_{r,j}\sigma_{r,i}\sigma_{r,j}x_ix_j\right)^2\right]$$

$$= \frac{2}{s^2}\mathbb{E}\left[\sum_{r=1}^{m}\sum_{i\neq j}\eta_{r,i}\eta_{r,j}x_i^2x_j^2\right]$$

$$\leq \frac{2}{m}\left(\sum_{i=1}^{n}x_i^2\right)^2$$

$$= \frac{2}{m}.$$

For $2 < q \leq m$, we apply Lemma $F.1$ and Lemma $F.2$. We only include $\alpha_4$ when $\ln(qmv^4/s^2) \geq 2$ to simplify the bound. The bound follows. $\square$

# G  Proof of Auxiliary Lemma for Lemma 2.4

We prove Lemma $E.2$.

*Proof of Lemma E.2.* First, we show the following fact: Suppose that there are $T$ distinguishable buckets and we want to a assign an ordered pair of 2 unequal elements in $[N]$ to each bucket so that the total number of times that any element $i \in [N]$ shows up is $\leq s$. We show that the number of such assignments is at least $C^T N^{2T}$ for some constant $C$. To prove this, we first consider the case where $N \geq 2T$. In this case, we have that the number of such assignments is at least:

$$N(N-1)(N-2)\ldots(N-2T+1) \geq C_1^{2T}N^{2T}.$$

Now, if $N < 2T$, then we define:

$$\beta = \left\lceil \frac{2T}{N} \right\rceil = \left\lceil 2Tv^2 \right\rceil \leq s.$$

We partition $2T$ into $\beta$ blocks, each of size $N$, until potentially the last block, which may be smaller. We can read off ordered pairs assigned to each bucket from this formulation. Let's assume that each block is a permutation of $1, \ldots, N$, and the last block is $2T - (\beta - 1)(N)$ non-equal numbers drawn from $1, \ldots, N$. (this satisfies the unequal ordered pair condition). Then the number of assignments is $(N!)^{\beta-1} \cdot (N)(N-1) \ldots (N - (2T - (\beta - 1)(N)) + 1)$. This is at least as big as $C_1^{2T} N^{2T}$ for some constant $C_1$.

First, we handle the case where $q/T = 2$. Since we have a uniform sparse JL distribution, we know that for $1 \leq x \leq s$:

$$\mathbb{E}[\eta_{1,1} \ldots \eta_{x,1}] \geq \frac{s(s-1) \ldots (s-x+1)}{(m)(m-1) \ldots (m-x+1)} \geq C_2^{-x} \left( \frac{s}{m} \right)^x.$$

We know that

$$Z_r^2 = 2 \left( \sum_{i \neq j} \eta_{r,i} \eta_{r,j} \right) + Y_r,$$

where $Y_r$ has expectation 0. In this case we have that

$$Z_1^2 \ldots Z_T^2 = 2^T \left( \sum_{i_1 \neq j_1, \ldots, i_T \neq j_T} \prod_{k=1}^{T} \eta_{k,i_k} \eta_{k,j_k} \right) + Q.$$

where $Q$ consists of terms that contain a factor of some $Y_r$. Due to the independence of the Rademachers, the expectation of any term that contains a factor of $Y_r$ has expectation 0, which implies that:

$$\mathbb{E}[Z_1^2 \ldots Z_T^2] = v^{2T} 2^T \mathbb{E} \left[ \left( \sum_{i_1 \neq j_1, \ldots, i_T \neq j_T} \prod_{k=1}^{T} \eta_{k,i_k} \eta_{k,j_k} \right) \right].$$

Let $\eta'_{r,i} \sim \eta_{r,i}$ be *independent* random variables. Suppose that

$$Z_r' := v^{2T} \sum_{i \neq j} \eta'_{r,i} \eta'_{r,j} \sigma_{r,i} \sigma_{r,j}.$$

We know that

$$Z_r'^2 = 2 \left( \sum_{i \neq j} \eta'_{r,i} \eta'_{r,j} \right) + Y_r',$$

where $Y_r'$ has expectation 0. This means that:

$$Z_1'^2 \ldots Z_T'^2 = v^{2T} 2^T \left( \sum_{i_1 \neq j_1, \ldots, i_T \neq j_T} \prod_{k=1}^{T} \eta'_{k,i_k} \eta'_{k,j_k} \right) + Q'$$

where $Q'$ consists of terms that contain a factor of some $Y_r'$. For similar reasons, this implies that

$$\mathbb{E}[Z_1'^2 \ldots Z_T'^2] = v^{2T} 2^T \mathbb{E} \left[ \left( \sum_{i_1 \neq j_1, \ldots, i_T \neq j_T} \prod_{k=1}^{T} \eta'_{k,i_k} \eta'_{k,j_k} \right) \right].$$

Let's view $\prod_{k=1}^{T} \eta'_{k,i_k} \eta'_{k,j_k}$ and $\prod_{k=1}^{T} \eta_{k,i_k} \eta_{k,j_k}$ as terms in a sum. In the second expression, every term has expectation $\left( \frac{s}{m} \right)^{2T}$, and there are at most $N^{2T}$ terms. In the first expression, if there are $> s$ copies of any $i_k$ value, then the expectation is 0. Otherwise, the expectation varies between $C_2^{-2T} \left( \frac{s}{m} \right)^{2T}$ and $\left( \frac{s}{m} \right)^{2T}$. By the counting argument at the beginning of the proof, we know that there are at least $C_1^{2T} N^{2T}$ terms. This implies that

$$\|Z_1 \ldots Z_T\|_2 \gtrsim C^T \|Z_1' \ldots Z_T'\|_2 = C^T \|Z_1'\|_2^T = C^T \|Z_1\|_2^T$$

as desired.

Now, we handle the case of the general $q/T$. Since we have a uniform sparse JL distribution, we know that for $1 \leq x \leq s$:

$$\mathbb{E}[\eta_{1,1} \ldots \eta_{x,1}] \geq \frac{s(s-1) \ldots (s-x+1)}{(m)(m-1) \ldots (m-x+1)} \geq C_2^{-x} \left(\frac{s}{m}\right)^x.$$

We know that

$$(Z_r)^{q/T} = 2^{q/T-1} \sum_{i \neq j} (\eta_{r,i} \eta_{r,j})^{q/T} + Y_r,$$

where $Y_r$ has expectation $\geq 0$. In this case we have that

$$(Z_1 \ldots Z_T)^{q/T} = 2^{q-T} \left( \sum_{i_1 \neq j_1, \ldots, i_T \neq j_T} \prod_{k=1}^{T} (\eta_{k,i_k} \eta_{k,j_k})^{q/T} \right) + Q.$$

where $Q$ has expectation $\geq 0$. This implies that:

$$\mathbb{E}[Z_1^{q/T} \ldots Z_T^{q/T}] \geq v^{2T} 2^{q-T} \mathbb{E}\left[ \left( \sum_{i_1 \neq j_1, \ldots, i_T \neq j_T} \prod_{k=1}^{T} (\eta_{k,i_k} \eta_{k,j_k})^{q/T} \right) \right].$$

Let $\eta'_{r,i} \sim \eta_{r,i}$ be *independent* random variables, and let $M'_r = \sum_{i=1}^{N} \eta'_{r,i}$. Suppose:

$$Z'_r := v^{2T} \sum_{i \neq j} \eta'_{r,i} \eta'_{r,j} \sigma_{r,i} \sigma_{r,j}.$$

We know that

$$(Z'_r I_{M'_r=2})^{q/T} = 2^{q/T-1} \sum_{i \neq j} \left( \eta'_{r,i} \eta'_{r,j} I_{M'_r=2} \right)^{q/T} + Y'_r,$$

where $Y'_r$ has expectation 0. In this case we have that

$$(Z'_1 I_{M'_1=2} \ldots Z'_T I_{M'_T=2})^{q/T} = 2^{q-T} \left( \sum_{i_1 \neq j_1, \ldots, i_T \neq j_T} \prod_{k=1}^{T} \left( \eta'_{k,i_k} \eta'_{k,j_k} I_{M'_k=2} \right)^{q/T} \right) + Q'.$$

where $Q'$ consists of terms that contain a factor of some $Y'_r$. For similar reasons to the above, we have that:

$$\mathbb{E}[Z_1^{'q} \ldots Z_T^{'q}] = v^{2T} 2^{q-T} \mathbb{E}\left[ \left( \sum_{i_1 \neq j_1, \ldots, i_T \neq j_T} \prod_{k=1}^{T} \left( \eta'_{k,i_k} \eta'_{k,j_k} I_{M'_k=2} \right)^{q/T} \right) \right].$$

Let's view $\prod_{k=1}^{T} (\eta_{k,i_k} \eta_{k,j_k})^{q/T}$ and $\prod_{k=1}^{T} \left( \eta'_{k,i_k} \eta'_{k,j_k} I_{M'_k=2} \right)^{q/T}$ as terms in a sum. In the second expression, every term has expectation $\leq \left(\frac{s}{m}\right)^{2T}$ (the indicator can only *reduce* the expectation), and there are at most $N^{2T}$ terms. In the first expression, if there are $> s$ copies of any $i_k$ value, then the expectation is 0. Otherwise, the expectation varies between $C_2^{-2T} \left(\frac{s}{m}\right)^{2T}$ and $\left(\frac{s}{m}\right)^{2T}$. By the counting argument, we know that there are at least $C_1^{-2T} N^{2T}$ terms. This implies that

$$\|Z_1 \ldots Z_T\|_{q/T} \gtrsim C^T \left\| Z'_1 I_{M'_1=2} \ldots Z'_T I_{M'_T=2} \right\|_{q/T} = C^T \left\| Z'_1 I_{M'_1} = 2 \right\|_{q/T}^T = C^T \left\| Z_1 I_{M_1=2} \right\|_{q/T}^T$$

as desired. □

## H    Proof of Lemma 3.1 and Lemma 3.2

Recall that our proof of Theorem 1.5 requires cleaner bounds on moments of $\|R(x_1, \ldots, x_n)\|_q$ that follow simplifying the bounds in Lemma 2.3 and Lemma 2.4 at the target values of $v$. The proofs of these lemmas boil down to function bounding and simplification.

## H.1 Proof of Lemma 3.1

First, we show how Lemma 2.3 implies Lemma 3.1. The proof involves simplifying and bounding the function at the target $v$ value.

*Proof of Lemma* 3.1. We plug $q = p$ into Lemma 2.3. We use this relaxed version of the bound: If $\frac{se}{mv^2} \geq q$, then $\|R(x_1, \ldots, x_n)\|_q \lesssim \frac{\sqrt{q}}{\sqrt{m}}$. Otherwise, if there exists $C_2 q^3 mv^4 \geq s^2$, then

$$
\|R(x_1, \ldots, x_n)\|_q \lesssim
\begin{cases}
\max\left( \frac{\sqrt{q}}{\sqrt{m}}, \frac{C_2^{1/3} q^2 v^2}{s \ln^2(qmv^2/s)} \right) & \text{if } \ln(qmv^4/s^2) \leq 2 \\[2ex]
\max\left( \frac{\sqrt{q}}{\sqrt{m}}, \frac{qv^2}{s \ln(qmv^4/s^2)}, \min\left( \frac{C_2^{1/3} q^2 v^2}{s \ln^2(qmv^2/s)}, \frac{q}{s \ln(m/s)} \right) \right) & \text{if } \ln(qmv^4/s^2) > 2
\end{cases}
$$

Suppose that the absolute constant on the upper bounds is $\leq C'$. Let $C = \max(C', 1)$ (we take $C$ to be the constant on the upper bounds). Let's take $C_{v,2} = \frac{0.25}{\sqrt{C}}$, $C_{v,1} = \min\left( \frac{0.1}{C^{3/2}}, C_{v,2} \right)$, $C_S = 4C$, $C_M = \max\left( e^{\frac{1}{C_{v,1}^2}}, 16C^2, e^{\frac{1}{C_{v,2}^2}}, e^2 \right)$. For the remainder of the analysis, we assume that $m \geq C_M \epsilon^{-2} p$ and $m < 2\epsilon^{-2}\delta$.

First, observe $m \geq 16C^2 \epsilon^{-2} p$ gives us that $C \frac{\sqrt{p}}{\sqrt{m}} \leq 0.25\epsilon$ regardless of $v$.

Now, let $f_1 = C_{v,1} \sqrt{\epsilon s} \frac{\ln(\frac{m\epsilon}{p})}{p}$, and $f_2 = \frac{C_{v,2} \sqrt{\epsilon s} \sqrt{\ln \frac{m\epsilon^2}{p}}}{\sqrt{p}}$.

First, let's analyze $v = f_2$. We show that $\ln(pmf_2^4/s^2) \geq 2$. Observe that $\ln(pmf_2^4/s^2) = \ln(C_{v,2}^4 \ln^2(m\epsilon^2/p)) + \ln(m\epsilon^2/p)$. Using the fact that $m \geq e^{\frac{1}{C_{v,2}^2}} \epsilon^{-2} p$, we see that

$$
C_{v,2}^4 \ln^2(m\epsilon^2/p) \geq C_{v,2}^4 \frac{1}{C_{v,2}^4} = 1.
$$

Now, since $m \geq e^2 \epsilon^{-2} p$, this implies that

$$
\ln(pmf_2^4/s^2) = \ln(C_{v,2}^4 \ln^2(m\epsilon^2/p)) + \ln(m\epsilon^2/p) \geq \ln(m\epsilon^2/p) \geq 2.
$$

Moreover, we know that $p^3 mf_2^2 \geq s^2$, since $pmv^4 \geq e^2 s^2$. Now, we show that $C \frac{pf_2^2}{s \ln(pmf_2^4/s^2)} \leq 0.25\epsilon$. Let's observe that

$$
\frac{Cpv^2}{s \ln(pmf_2^4/s^2)} \leq \frac{CC_{v,2}^2 \epsilon \ln(\frac{m\epsilon^2}{p})}{\ln(\frac{m\epsilon^2}{p})} = CC_{v,2}^2 \epsilon
$$

Since $C_{v,2} = \frac{0.25}{\sqrt{C}}$, we get a bound of $0.25\epsilon$.

Now, we handle the case where $m \geq s \cdot e^{\frac{C_S p \epsilon^{-1}}{s}}$. We first show that $C \frac{p}{s \ln(m/s)} \leq 0.25\epsilon$. If $s \geq \Theta(\epsilon^{-1} \ln(1/\delta))$, using that $m \geq se$, this immediately follows from $\frac{p}{s \ln(m/s)} \leq \frac{p}{s} \leq 0.25\epsilon$. Otherwise, we need it to be true that $s \ln(m/s) \geq 4Cp\epsilon^{-1}$. This can be written as $\ln(m/s) \geq \frac{4Cp\epsilon^{-1}}{s}$. Since $C_S = 4C$, this can be written as: $m \geq s \cdot e^{\frac{C_S p \epsilon^{-1}}{s}}$, as desired. This, combined with the above analysis, implies that when $m \geq s \cdot e^{\frac{C_S p \epsilon^{-1}}{s}}$, taking $v = f_2$:

$$
\|R(x_1, \ldots, x_n)\|_q \leq C \max\left( \frac{\sqrt{q}}{\sqrt{m}}, \frac{qv^2}{s \ln(qmv^4/s^2)}, \min\left( \frac{C_2^{1/3} q^2 v^2}{s \ln^2(qmv^2/s)}, \frac{q}{s \ln(m/s)} \right) \right)
$$

$$
\leq C \max\left( \frac{\sqrt{q}}{\sqrt{m}}, \frac{qv^2}{s \ln(qmv^4/s^2)}, \frac{q}{s \ln(m/s)} \right)
$$

$$
\leq 0.25\epsilon.
$$

Now, we just need to handle the case where $m \leq s \cdot e^{C_S p \epsilon^{-1}/s}$, $m \geq \Theta(\epsilon^{-1} \ln(1/\delta))$, $m \geq se$. Such values only exist if $s \leq \Theta(\epsilon^{-1} \ln(1/\delta))$. Observe that we can set $C_2 = \frac{1}{C_{v,1}^4}$ and using the fact that $C_{v,1} \leq C_{v,2}$, we obtain that

$$
\frac{C_2 p^3 mv^4}{s^2} \geq \frac{C_2 p^3 m}{s^2} \min(C_{v,1}, C_{v,2})^4 \left( \frac{\sqrt{\epsilon s}}{p} \right)^4 = C_2 C_{v,1}^4 \frac{m\epsilon^2}{p} \geq C_2 C_{v,1}^4.
$$

Thus, this is lower bounded by 1 when $C_2 = \frac{1}{C_{v,1}^4}$.

First, we analyze the case of $v = f_1$. We show that $\frac{CC_2^{1/3}p^2v^2}{s\ln^2(pmv^2/s)} \le 0.1\epsilon$. Observe that

$$\frac{CC_2^{1/3}p^2v^2}{s\ln^2(pmv^2/s)} = \frac{\epsilon CC_2^{1/3}C_{v,1}^2\ln^2(\frac{m\epsilon}{p})}{\ln^2(\frac{C_{v,1}^2 m\epsilon\ln^2(\frac{m\epsilon}{p})}{p})} = \frac{\epsilon CC_2^{1/3}C_{v,1}^2\ln^2(\frac{m\epsilon}{p})}{\left(\ln(\frac{m\epsilon}{p}) + \ln(C_{v,1}^2\ln^2(\frac{m\epsilon}{p}))\right)^2}.$$

Now, since $m \ge e^{1/C_{v,1}^2}\epsilon^{-2}p$, we know that $\ln(C_{v,1}^2\ln^2(\frac{m\epsilon}{p})) \ge 0$. Thus we can bound the above expression by:

$$\frac{\epsilon CC_{v,1}^{2/3}\ln^2(\frac{m\epsilon}{p})}{\ln^2(\frac{m\epsilon}{p})} = \epsilon CC_{v,1}^{2/3}\epsilon \le 0.1\epsilon,$$

where the last inequality uses the fact that $C_{v,1} \le \frac{0.1}{C^{3/2}}$.

Let's now consider how the term $\frac{pv^2}{s\ln(pmv^4/s^2)}$ how changes as a function of $v$. This term only arises in the bound if $\ln(pmv^4/s^2) \ge 2$. First, we show this is an increasing function of $v$. Let $w = pmv^4/s^2$. We see that $\frac{pv^2}{s\ln(pmv^4/s^2)} = \frac{s}{\sqrt{pm}}\frac{\sqrt{w}}{\ln w}$. We observe that this is an increasing function of $w$ as long as $w \ge e^2$, which is exactly our restriction on $w$. Thus, $\frac{pv^2}{s\ln(pmv^4/s^2)}$ is an increasing function of $v$ in this range.

Now, we consider how the $\frac{C_2^{1/3}p^2v^2}{s\ln^2(pmv^2/s)}$ term changes a function of $v$. This term only arises in the bound if $\ln(pmv^2/s) \ge 1$. First, we show that $f(v) \le 2f(v')$ if $v \le v'$. Let $w = pmv^2/s$. We see that $\frac{p^2v^2}{s\ln(pmv^2/s)} = \frac{p}{m}\frac{w}{\ln^2 w}$. We observe that this is an increasing function of $w$ as long as $w \ge e^2$. When $e \le w \le e^2$, observe that this is bounded by at most a factor of 2 above any other $w$ value.

Now, for the remainder of the analysis, let $v = \min(f_1, f_2)$. We show that $\|R(x_1, \ldots, x_n)\|_q \le 0.25\epsilon$.

If $\ln(pmv^2/s) \le 1$ (i.e. $\frac{se}{mv^2} \ge p$), then we know that the bound is actually $\frac{\sqrt{p}}{\sqrt{m}}$, and we've already shown that $\|R(x_1, \ldots, x_n)\|_q \le 0.25\epsilon$.

For the remainder of the analysis, we assume that $\ln(pmv^2/s) > 1$.

First, suppose that $v = f_1$. If $\ln(pmv^4/s^2) \le 2$, then we know that

$$\|R(x_1, \ldots, x_n)\|_q \le C\max\left(\frac{\sqrt{q}}{\sqrt{m}}, \frac{C_2^{1/3}q^2v^2}{s\ln^2(qmv^2/s)}\right) \le 0.25\epsilon.$$

Otherwise, we know that $\ln(pmv^4/s^2) > 2$. First let's show that that $C\frac{pv^2}{s\ln(pmv^4/s^2)} \le 0.25\epsilon$. We know that $v \le f_2$. At $v = f_2$, we know that the expression is upper bounded by $0.25\epsilon$. Since the $\frac{pv^2}{s\ln(pmv^4/s^2)}$ term is an increasing function of $v$ in this regime, this means that we get a bound of $0.25\epsilon$ in this case too. Thus, we know that:

$$\|R(x_1, \ldots, x_n)\|_q \le C\max\left(\frac{\sqrt{q}}{\sqrt{m}}, \frac{qv^2}{s\ln(qmv^4/s^2)}, \min\left(\frac{C_2^{1/3}q^2v^2}{s\ln^2(qmv^2/s)}, \frac{q}{s\ln(m/s)}\right)\right)$$

$$\le C\max\left(\frac{\sqrt{q}}{\sqrt{m}}, \frac{qv^2}{s\ln(qmv^4/s^2)}, \frac{C_2^{1/3}q^2v^2}{s\ln^2(qmv^2/s)}\right)$$

$$\le 0.25\epsilon.$$

Now, suppose that $v = f_2$. We've already shown that $\ln(pmv^4/s^2) \ge 2$ here (near the beginning of the proof). Since $v \le f_1$, we obtain a bound of $2 \cdot 0.1\epsilon = 0.2\epsilon$. This means:

$$\|R(x_1, \ldots, x_n)\|_q \le C\max\left(\frac{\sqrt{q}}{\sqrt{m}}, \frac{qv^2}{s\ln(qmv^4/s^2)}, \min\left(\frac{C_2^{1/3}q^2v^2}{s\ln^2(qmv^2/s)}, \frac{q}{s\ln(m/s)}\right)\right)$$

$$\le C\max\left(\frac{\sqrt{q}}{\sqrt{m}}, \frac{qv^2}{s\ln(qmv^4/s^2)}, \frac{C_2^{1/3}q^2v^2}{s\ln^2(qmv^2/s)}\right)$$

$$\le 0.25\epsilon.$$

$\square$

## H.2 Proof of Lemma 3.2

Now, we show how Lemma 2.3 and Lemma 2.4 imply Lemma 3.2. The proof simply involves bounding and simplifying the functions in the original lemmas at the target $v$ value.

*Proof of Lemma* 3.2. We use Lemma 2.3 but put in an absolute constant. Let $D_2 > 0$ be such that: if $\frac{se}{mv^2} \geq q$, then

$$\|R(x_1, \ldots, x_n)\|_q \leq D_2 \frac{\sqrt{q}}{\sqrt{m}}.$$

Otherwise, if $q^3 m v^4 \geq s^2$, then $\|R(x_1, \ldots, x_n)\|_q$ is upper bounded by:

$$D_2 \begin{cases} \max\left(\frac{\sqrt{q}}{\sqrt{m}}, \frac{q^2 v^2}{s \ln^2(qmv^2/s)}\right) & \text{if } \ln(qmv^4/s^2) \leq 2, \ln(qmv^2/s) \leq q \\ \frac{\sqrt{q}}{\sqrt{m}} & \text{if } \ln(qmv^4/s^2) \leq 2, \ln(qmv^2/s) > q \\ \max\left(\frac{\sqrt{q}}{\sqrt{m}}, \frac{4096 q v^2}{s \ln(qmv^4/s^2)}, \min\left(\frac{q^2 v^2}{s \ln^2(qmv^2/s)}, \frac{q}{s \ln(m/s)}\right)\right) & \text{if } \ln(qmv^4/s^2) > 2, \ln(qmv^2/s) \leq q \\ \max\left(\frac{\sqrt{q}}{\sqrt{m}}, \frac{4096 q v^2}{s \ln(qmv^4/s^2)}\right) & \text{if } \ln(qmv^4/s^2) > 2, \ln(qmv^2/s) > q. \end{cases}$$

We use Lemma 2.4 but put in an absolute constant $D_1 > 0$ (which we take to be $\leq 1$). Let $2 \leq q \leq m$ be an even integer, and suppose that $0 < v \leq 0.5$ and $\frac{1}{v^2}$ is an even integer. If $qv^2 \leq s$, then

$$\|R(x_1, \ldots, x_n)\|_q \geq D_1 \frac{\sqrt{q}}{\sqrt{m}}.$$

If $m \geq q$, $2 \leq \ln(qmv^4/s^2) \leq q$, $2qv^2 \leq 0.5s \ln(qmv^4/s^2)$, and $s \leq m/e$ then:

$$\|R(x_1, \ldots, x_n)\|_q \geq D_1 \frac{4096 q v^2}{s \ln(qmv^4/s^2)}.$$

If $v \leq \frac{\sqrt{\ln(m/s)}}{\sqrt{q}}$ and $1 \leq \ln(qmv^2/s) \leq q/2$, and $s \leq m/e$, then:

$$\|R(x_1, \ldots, x_n)\|_q \geq D_1 \frac{q^2 v^2}{s \ln^2(qmv^2/s)}.$$

Let $D = \frac{D_1}{2048 D_2}$. It suffices to show that for $v$ defined in the lemma statement, $\|R(v, v, \ldots, v, 0, \ldots, 0)\|_q \geq 2\epsilon$ and

$$\frac{\|R(v, v, \ldots, v, 0, \ldots, 0)\|_q}{\|R(v, v, \ldots, v, 0, \ldots, 0)\|_{2q}} \geq D.$$

First, we handle the case where $m \leq \Theta(\epsilon^{-2} \ln(1/\delta))$. Let's take $v = \psi$ for any sufficiently small $\psi$. By sufficiently small, we mean $v^2 \leq \frac{se}{2mq}$ and $0 < v \leq 0.5$. This implies that $\frac{se}{mv^2} \geq 2q$ and $qv^2 \leq s$. Thus we know (using that $q \leq m$) that $\|R(x_1, \ldots, x_n)\|_{2q} \leq D_2 \frac{\sqrt{2q}}{\sqrt{m}}$ and $\|R(x_1, \ldots, x_n)\|_q \geq D_1 \frac{\sqrt{q}}{\sqrt{m}}$. This means that:

$$\frac{\|R(v, v, \ldots, v, 0, \ldots, 0)\|_q}{\|R(v, v, \ldots, v, 0, \ldots, 0)\|_{2q}} \geq D$$

as desired. Suppose that $m \leq \Theta(\epsilon^{-2} \ln(1/\delta))$. Based on the setting $q$, this means that $\|R(v, \ldots, v, 0 \ldots, 0)\|_q \geq D_1 \frac{\sqrt{q}}{\sqrt{m}} \geq 2\epsilon$ as desired.

Now, we handle the cases where $m \geq \Theta(\epsilon^{-2} \ln(1/\delta))$. Notice that the condition $f'(m, \epsilon, \delta, s) \leq 0.5$ allows us to assume that $s \leq \Theta(\epsilon^{-1} \ln(1/\delta))$ and $m \leq \epsilon^{-2} e^p$. Let $f_1 = 4\sqrt{\epsilon s} \frac{\ln(\frac{m\epsilon}{q})}{q}$ and let $f_2 = \sqrt{\epsilon s} \frac{\sqrt{\ln(\frac{m\epsilon^2}{q})}}{\sqrt{q}}$. We will consider $v = C_{v,1} f_1 =: v_1$ and $v = C_{v,2} f_2 =: v_2$. First, we handle the condition of $q^3 m v^4 \geq s^2$. We enforce the condition $C_{v,1}, C_{v,2} \geq 1$. Assuming that $v \geq \frac{\sqrt{\epsilon s}}{q}$ (which is true at the two values of $v$ that we consider), we know $\frac{q^3 m v^4}{s^2} \geq \frac{m\epsilon^2}{q} \geq 1$. Also, we make $m \geq 2C^2 \epsilon^{-2} q$, so that $\sqrt{\frac{2q}{m}} \leq \sqrt{\frac{2q}{2C^2 \epsilon^{-2} q}} = \frac{\epsilon}{C}$.

Consider $v = v_2$. We first check that the conditions for the upper bound are satisfied. We have that $\frac{qmv^4}{s^2} = C_{v,2}^4 \frac{m\epsilon^2}{q} \ln^2(\frac{m\epsilon^2}{q})$. Observe that when $m \geq e^2 \epsilon^{-2} q$ and $C_{v,2} \geq 1$, this is lower bounded by $e^2$, so

$\ln(qmv^4/s^2) \geq 2$. Also, we have that $\frac{qmv^2}{se} = \sqrt{qm}\sqrt{\frac{qmv^4}{s^2}}\frac{1}{e} \geq 1$. Now, we check the additional conditions needed for the lower bound. Observe that

$$\frac{2qv^2}{s} = 2\epsilon C_{v,2}^2 \ln(\frac{m\epsilon^2}{q}) \leq 0.5 C_{v,2}^4 \frac{m\epsilon^2}{q} \ln^2(\frac{m\epsilon^2}{q}) = 0.5\ln(qmv^4/s^2)$$

as desired. We check that $\ln(qmv^4/s^2) \leq q$. It suffices to show that

$$\frac{m\epsilon^2}{q}\ln^2\left(\frac{m\epsilon^2}{q}\right) \leq \frac{e^q}{C_{v,2}^4}.$$

Using the condition that $m \leq \epsilon^{-2}\frac{e^q}{qC_{v,2}^4}$ where we obtain that

$$\frac{m\epsilon^2}{q}\ln^2(\frac{m\epsilon^2}{q}) \leq \frac{e^q}{q^2 C_{v,2}^4}\ln^2(\frac{e^q}{q^2 C_{v,2}^4}) \leq \frac{e^q}{q^2 C_{v,2}^4}\ln^2(e^q) \leq \frac{e^q}{C_{v,2}^4}$$

as desired. Now, we compute the value of $\frac{qv^2}{s\ln(qmv^4/s^2)}$ at $v = C_{v,2}f_2$. We obtain:

$$\frac{qv^2}{s\ln(qmv^4/s^2)} = C_{v,2}^2\epsilon \frac{\ln(m\epsilon^2/q)}{\ln\left(\frac{m\epsilon^2}{q}\right) + \ln\left(C_{v,2}^4\ln^2\left(\frac{m\epsilon^2}{q}\right)\right)}.$$

Consider $v = v_1$. We first check that the conditions for the upper bound are satisfied. In this case, we have that $\frac{qmv^2}{s} = 16C_{v,1}^2\frac{m\epsilon}{q}\ln^2(\frac{m\epsilon}{q})$. Observe that when $C_{v,1} \geq 1$ and $m \geq e^2\epsilon^{-2}q \geq e^2\epsilon^{-1}q$, this is lower bounded by $e^2$, so $\ln(qmv^2/s) \geq 2$. Now, we claim that when $f_1 \leq f_2$, we show that $\ln(qmv^2/s) \leq q/2$. In this case, using that $m \leq \epsilon^{-2}qe^q$, we have: $\frac{4\ln(m\epsilon/q)}{q} \leq \frac{\sqrt{\ln(m\epsilon^2/q)}}{\sqrt{q}}$. This means that $\ln(m\epsilon/q) \leq \sqrt{q}\sqrt{\ln(m\epsilon^2/q)}/4 \leq q/4$. Observe that

$$\ln(qmv^2/s) = \ln(16C_{v,1}^2) + \ln(m\epsilon/q) + 2\ln\ln(m\epsilon/q)$$
$$\leq \ln(16C_v^2) + \frac{q}{4} + 2\ln\ln q$$
$$\leq \frac{q}{2}.$$

At this value, observe that:

$$\frac{q^2v^2}{s\ln^2(qmv^2/s)} = 16C_{v,1}^2\epsilon\left(\frac{\ln(m\epsilon/q)}{\ln\left(\frac{m\epsilon}{q}\right) + \ln\left(16C_{v,1}^2\ln^2\left(\frac{m\epsilon}{q}\right)\right)}\right)^2.$$

Let $C = D_1$. Let's set $\sqrt{\frac{1}{C}} \leq C_{v,2} = C_{v,1} = C_v \leq 4\sqrt{\frac{1}{C}}$. Using the fact that $v^2 \leq 0.5$ (so $\frac{1}{v^2} \geq 2$), this means that $\frac{1}{v^2}$ has can take on at least 3 different powers of 2. Let's observe that when $16C_v^2\ln^2(\frac{m\epsilon}{q}) \leq \frac{m\epsilon}{q}$ (we can get this condition by saying that $m \geq C_{M,2}\epsilon^{-2}q$ for a sufficiently large $C_{M,2}$) and $16C_v^2\ln^2(m\epsilon/q) \geq 1$ (we can get this condition by saying that $m \geq C_{M,2}\epsilon^{-2}q$ for a sufficiently large $C_{M,2}$), we know that

$$\frac{4\epsilon}{C} \leq 4C_v^2\epsilon \leq \frac{q^2v_1^2}{s\ln^2(qmv_1^2/s)} \leq 16C_v^2\epsilon \leq \frac{256\epsilon}{C}.$$

Suppose that $C_v^4\ln^2(m\epsilon^2/q) \leq \frac{m\epsilon^2}{q}$ (we can get this condition by saying that $m \geq C_{M,2}\epsilon^{-2}q$ for a sufficiently large $C_{M,2}$) and $C_v^4\ln^2(m\epsilon^2/q) \geq 1$ (we can get this condition by saying that $m \geq C_{M,2}\epsilon^{-2}q$ for a sufficiently large $C_{M,2}$). Let's observe that

$$4096C_v^2\epsilon \geq \frac{4096qv_2^2}{s\ln^2(qmv_2^4/s^2)} \geq 2048C_v^2\epsilon \geq \frac{2048\epsilon}{C}.$$

Let $m' = s \cdot e^{\frac{C\epsilon^{-1}q}{1024s}}$. When $m \geq m'$, we know that $\frac{q}{s\ln(m/s)} \leq \frac{1024\epsilon}{C}$ and when $m \leq m'$, we know that $\frac{q}{s\ln(m/s)} \geq \frac{1024\epsilon}{C}$.

In order to plug in $v = v_1$ and use the $\frac{q^2v^2}{s\ln^2(qmv^2/s)}$ lower bound, we need to show that $v \leq \frac{\sqrt{\ln(m/s)}}{\sqrt{q}}$. At $v = \frac{\sqrt{\ln(m/s)}}{\sqrt{q}}$, we have that $\frac{qmv^2}{s} = \frac{m}{s}\ln\left(\frac{m}{s}\right)$. Observe that when $m \geq e^2 s$, this is lower bounded by $e^2$, so $\ln(qmv^2/s) \geq 2$. At this value, observe that:

$$\frac{q^2v^2}{s\ln^2(qmv^2/s)} = \frac{q\ln(m/s)}{s\ln^2\left(\frac{m}{s}\ln\left(\frac{m}{s}\right)\right)} \geq \frac{q\ln(m/s)}{4s\ln^2\left(\frac{m}{s}\right)} = \frac{q}{4s\ln\left(\frac{m}{s}\right)}.$$

We can write $\frac{q^2v^2}{s\ln^2(qmv^2/s)} = \frac{q}{m}\frac{w}{\ln^2 w}$, where $w = qmv^2/s$. We observe that this is an increasing function of $w$ as long as $w \geq e^2$. Thus, it suffices to show that $\frac{q^2v_1^2}{s\ln^2(qmv_1^2/s)} \leq \frac{q^2v^2}{s\ln^2(qmv^2/s)}$. When $m \leq m'$, we know that

$$\frac{q^2v_1^2}{s\ln^2(qmv_1^2/s)} \leq \frac{256\epsilon}{C} \leq \frac{q}{4s\ln\left(\frac{m}{s}\right)} \leq \frac{q^2v^2}{s\ln^2(qmv^2/s)}.$$

Thus, we have that $v_1 \leq v = \frac{\sqrt{\ln(m/s)}}{\sqrt{q}}$ as desired.

The first case is $m \leq m'$ and $f_2 \leq f_1$. We set $v = C_v f_2$.

$$\|R(v,\ldots,v,0,\ldots,0)\|_q \geq D_1 \frac{4096qv^2}{s\ln(qmv^4/s)}.$$

For the upper bound, we see that $\ln(2qmv^4/s^2) > \ln(qmv^4/s^2) \geq 2$ and $\sqrt{\frac{2q}{m}} \leq \frac{\epsilon}{C}$. Here, we have that

$$\|R(v,\ldots,v,0,\ldots,0)\|_{2q} \leq \begin{cases} D_2 \max\left(\frac{\sqrt{2q}}{\sqrt{m}}, \frac{8192qv^2}{s\ln(2qmv^4/s)}, \frac{4q^2v^2}{s\ln^2(2qmv^2/s)}\right) & \text{if } \ln(2qmv^2/s) \leq 2q \\ D_2 \max\left(\frac{\sqrt{2q}}{\sqrt{m}}, \frac{8192qv^2}{s\ln(qmv^4/s)}\right) & \text{if } \ln(2qmv^2/s) > 2q \end{cases}.$$

Now, we use the fact that $v \leq C_v f_1 := v_1$ to see that:

$$\frac{4q^2v^2}{s\ln(2qmv^2/s)} \leq \frac{4q^2v^2}{s\ln(qmv^2/s)} \leq \frac{8q^2v_1^2}{s\ln(qmv_1^2/s)} \leq \frac{2048\epsilon}{C}.$$

We also observe that since $2qmv^4/s \leq (qmv^4/s)^2$, we know:

$$\frac{8192qv^2}{s\ln(2qmv^4/s)} \geq \frac{8192qv^2}{2s\ln(qmv^4/s)} \geq \frac{2048\epsilon}{C}.$$

This, coupled with the guarantee on $\frac{\sqrt{2q}}{\sqrt{m}}$, implies we have an upper bound of:

$$\|R(v,\ldots,v,0,\ldots,0)\|_{2q} \leq D_2 \frac{8192qv^2}{s\ln(2qmv^4/s)}.$$

Thus, we have that

$$\frac{\|R(v,\ldots,v,0\ldots,0)\|_q}{\|R(v,\ldots,v,0,\ldots,0)\|_{2q}} \geq \frac{D_1}{2D_2} \geq D.$$

Moreover, we have that

$$\|R(v,\ldots,v,0,\ldots,0)\|_q \geq D_1 \cdot \frac{4096qv^2}{s\ln(qmv^4/s)} \geq D_1 \frac{2048\epsilon}{C} = 2048\epsilon$$

The next case is $f_1 \leq f_2$ and $m \leq m'$. We set $v = v_1$. Since $f_1 \leq f_2$, we know that $\ln(qmv^4/s^2) \leq q$. Thus we know:

$$\|R(v,\ldots,v,0,\ldots,0)\|_q \geq \begin{cases} D_1 \max\left(\frac{4096qv^2}{s\ln(qmv^4/s)}, \frac{q^2v^2}{s\ln^2(qmv^2/s)}\right) & \text{if } \ln(qmv^4/s^2) \geq 2, qv^2 \leq s \\ D_1 \frac{q^2v^2}{s\ln^2(qmv^2/s)} & \text{else} \end{cases}.$$

For the upper bound, we know that:

$$\|R(v,\ldots,v,0,\ldots,0)\|_{2q} \leq \begin{cases} D_2 \max\left(\frac{\sqrt{2q}}{\sqrt{m}}, \frac{8192qv^2}{s\ln(2qmv^4/s)}, \frac{4q^2v^2}{s\ln^2(2qmv^2/s)}\right) & \text{if } \ln(2qmv^4/s) > 2 \\ D_2 \max\left(\frac{\sqrt{2q}}{\sqrt{m}}, \frac{4q^2v^2}{s\ln^2(2qmv^2/s)}\right) & \text{if } \ln(2qmv^4/s) \leq 2 \end{cases}.$$

To make these bounds compatible, we need to handle the case where $\ln(qmv^4/s) \geq 2$, $qv^2 \geq s$ better. Let $v' = C_v f_2$. Assuming that $\ln(qmv^4/s) \geq 2$, we know that $\frac{8192qv^2}{s\ln(2qmv^4/s)}$ can be upper bounded by:

$$\frac{8192qv^2}{s\ln(qmv^4/s)} \leq \frac{8192qv'^2}{s\ln(qmv'^4/s)} = \frac{8192C_v^2\epsilon\ln(m\epsilon^2/q)}{\ln(m\epsilon^2/q) + \ln(C_v^4\ln^2(m\epsilon^2/q))} \leq 8192C_v^2\epsilon \leq \frac{8192\epsilon}{C}$$

as long as $\ln^2(m\epsilon/q)C_v^4 \geq 1$ (which we can make true by appropriately setting the constants on the bound for $m$). Observe also that:

$$\frac{4q^2v^2}{s\ln^2(2qmv^2/s)} \geq \frac{q^2v^2}{s\ln^2(qmv^2/s)} \geq \frac{4\epsilon}{C}.$$

Thus:

$$\frac{8192qv^2}{s\ln(qmv^4/s)} \leq \frac{8192q^2v^2}{s\ln^2(2qmv^2/s)}.$$

This, coupled with the guarantee on $\frac{\sqrt{2q}}{\sqrt{m}}$, implies that our upper bound becomes:

$$\|R(v,\ldots,v,0,\ldots,0)\|_{2q} \leq \begin{cases} D_2\frac{8192q^2v^2}{s\ln^2(2qmv^2/s)} & \text{if } \ln(2qmv^4/s) \leq 2 \text{ or } \ln(qmv^4/s) \geq 2, qv^2 \geq s \\ D_2 \max\left(\frac{8192qv^2}{s\ln(2qmv^4/s)}, \frac{4q^2v^2}{s\ln^2(2qmv^2/s)}\right) & \text{else }. \end{cases}$$

We now show that we can tweak $C_v$ within the factor of $2^{1/4}$ range permitted to show that we can ensure that it is not true that $2 - \ln 2 < \ln(qmv^4/s) \leq 2$. Observe that multiplying by a factor of $2^{1/4}$ in this case yields $\ln(2qmv^4/s) > 2$ and dividing by a factor of $2^{1/4}$ yields $\ln(qmv^4/s) \leq 2 - \ln 2$. Thus, at least one of the $C_v$ values that yields a power of 2 for $\frac{1}{v^2}$ will work. Thus, we have that

$$\frac{\|R(v,\ldots,v,0\ldots,0)\|_q}{\|R(v,\ldots,v,0,\ldots,0)\|_{2q}} \geq \frac{D_1}{8192D_2} = \frac{D}{2048}.$$

Moreover, we have that:

$$\|R(v,\ldots,v,0,\ldots,0)\|_q \geq D_1 \cdot \frac{q^2v^2}{s\ln^2(qmv^2/s)} \geq D_1\frac{4\epsilon}{C} = 4\epsilon$$

The next case is that $m > m'$. We set $v = C_v\sqrt{\epsilon s}\frac{\sqrt{\ln\left(\frac{m\epsilon^2}{q}\right)}}{\sqrt{q}}$. We know:

$$\|R(v,\ldots,v,0,\ldots,0)\|_q \geq D_1\frac{4096qv^2}{s\ln(qmv^4/s)}.$$

For the upper bound, we see that $\ln(2qmv^4/s^2) > \ln(qmv^4/s^2) > 2$. We know:

$$\|R(v,\ldots,v,0,\ldots,0)\|_{2q} \leq \begin{cases} D_2 \max\left(\frac{\sqrt{2q}}{\sqrt{m}}, \frac{8192qv^2}{s\ln(2qmv^4/s)}, \frac{2q}{s\ln(m/s)}\right) & \text{if } \ln(2qmv^2/s) \leq 2q \\ D_2 \max\left(\frac{\sqrt{2q}}{\sqrt{m}}, \frac{8192qv^2}{s\ln(2qmv^4/s)}\right) & \text{if } \ln(2qmv^2/s) > 2q \end{cases}.$$

This can be relaxed to:

$$\|R(v,\ldots,v,0,\ldots,0)\|_{2q} \leq D_2 \max\left(\frac{\sqrt{2q}}{\sqrt{m}}, \frac{8192qv^2}{s\ln(2qmv^4/s)}, \frac{2q}{s\ln(m/s)}\right).$$

Now, we know that

$$\frac{2q}{s\ln(m/s)} \leq \frac{2048\epsilon}{C} \leq \frac{4096qv^2}{s\ln(qmv^4/s)} = \frac{8192qv^2}{2s\ln(qmv^4/s)} \leq \frac{8192qv^2}{s\ln(2qmv^4/s)}.$$

This coupled with what we know about $\frac{\sqrt{2q}}{\sqrt{m}}$ means that:

$$\|R(v,\ldots,v,0,\ldots,0)\|_{2q} \leq D_2 \frac{8192qv^2}{s\ln(2qmv^4/s)}.$$

Thus, we have that

$$\frac{\|R(v,\ldots,v,0\ldots,0)\|_q}{\|R(v,\ldots,v,0,\ldots,0)\|_{2q}} \geq \frac{D_1}{2D_2} \geq D.$$

Moreover, we have that

$$\|R(v,\ldots,v,0,\ldots,0)\|_q \geq D_1 \cdot \frac{4096qv^2}{s\ln(qmv^4/s)} \geq D_1 \frac{2048\epsilon}{C} = 2048\epsilon.$$

We use the condition on $q$ not being more than a constant factor away from $p = \ln(1/\delta)$, to conclude that $\epsilon^{-2}q = \Theta(\epsilon^{-2}p)$, $f_2 = \Theta\left(\sqrt{\epsilon s}\frac{\sqrt{\ln\left(\frac{m\epsilon^2}{p}\right)}}{\sqrt{p}}\right)$, and $f_1 = \Theta\left(\sqrt{\epsilon s}\frac{\ln\left(\frac{m\epsilon}{p}\right)}{p}\right)$, and to conclude that the boundaries move within the $\Theta$ notation as well. $\qquad\square$

# I   Additional Experimental Results and Discussion

All of the experiments (in the main paper and the Appendix) were run on the default hardware on a Google Colab notebook. The code is available at `https://github.com/mjagadeesan/sparsejl-featurehashing`.

First, we give the results of additional experimental results on real-world and synthetic datasets, using the same experimental setup as the main paper.

Figure 5: Phase transitions of $\hat{v}(m, 0.2, 0.01, s)$

Figure 6: Phase transitions of $\hat{v}(m, 0.02, 0.05, s)$

For the synthetic datasets, the trends in Figure 5 and Figure 6 look quite similar to the figures in the main paper. We see, though, that Figure 6 experiences more severe non-monotonic behavior as a function of $s$ in the second phase transition. Consider, for example, in Figure 6, the behavior at $m = 12000$: we see that $\hat{v}(m, \epsilon, \delta, 4) < \hat{v}(m, \epsilon, \delta, 3)$. In fact, the order of the phase transitions in Figure 6 is far from decreasing. Nonetheless, the general patterns and trends in the theoretical result still hold (e.g. the "flat" part occurs at a lower y-coordinate for lower $s$ values.)

For the real-world datasets, the trends in Figure 7, Figure 8, and Figure 9 look quite similar to the figures in the main paper. One slight difference is that the failure probability noticeably increases in Figure 7 and Figure 8 between $s = 8$ and $s = 16$. It turns out that the failure probability actually increases to a local maximum somewhere in $12 \leq s \leq 16$, and then decreases when $s \geq 16$, reaching lower than the value at $s = 8$ by the time $s = 20$. There turns out to be a similar local maximum phenomenon when $\epsilon = 0.07$ and $m = 500$, though the local maximum occurs in $24 \leq s \leq 32$ and thus is not as visible in the graph.

As a general comment on non-monotonicity as a function of $s$, we emphasize that our asymptotic theoretical results characterize the *macroscopic* behavior of $v(m, \epsilon, \delta, s)$, and do not preclude the existence of constant factor fluctuations for small changes in parameters. An interesting direction for future work would be to look further into this non-mononocity and try to characterize when it arises.

Figure 7: $\hat{\delta}(m, s, 0.1)$ on News20

Figure 8: $\hat{\delta}(m, s, 0.1)$ on Enron

Figure 9: $\hat{\delta}(m, s, 0.03)$ on News20

## Footnotes

[1]In fact, $v = \frac{\sqrt{\ln(m/s)}}{\sqrt{T}}$ is very close to the value where $Tv^2 = \ln(Tmv^2/s)$, so this approximation is essentially tight.

[2]Observe that the upper endpoint of $T$ on the sup expression does not match with the upper endpoint of $T/2$ on the sup expression in Lemma $D$.1, and in fact, it turns out that this bound is not sufficiently strong to recover Theorem 1.5. This is sufficiently tight here, since we are focusing on the case where $\ln(Tmv'^2/s)$ is *small*.

[3]This can easily be seen by expanding.