[Reviews · NeurIPS 2019]

Reviewer 1



Note that a denser version of feature hashing was already considered in [29], under the name "Multiple Hashing". This is an alternative to sparse JL, seen as a preprocessing step that expands each dimension to s dimensions (scaling by 1/sqrt(s)) before applying feature hashing. This defines essentially the same mapping (only difference is whether a set or multiset of s dimensions is selected). Note that this decreases the L_infty/L_2 ratio by exactly a factor sqrt(s), so combining this with [23] yields lower bounds f(m,eps,p) that are exactly sqrt(s) times the bounds in [23] and not far from your tight bounds. This consequence of previous work should be acknowledged in your paper. Even though the L_infty/L_2 ratio has been used in the past to give bounds on the performance, it does not *characterise* the performance of feature hashing. For example, feature hashing handles a single large entry in a vector well, even though it may send L_infty/L_2 through the roof. It would be nice to have a more fine-grained understanding that has predictive power on any given vector.

Reviewer 2



This paper follows a recent line of research on analyzing sparse JL transform for vectors with bounded l_{\infty} norm to l_2 norm ratio. The current paper focuses on sparse JL transform with general sparsity parameter s and give tight bounds on tradeoff between l_{\infty} norm to l_2 norm ratio, accuracy and sparsity s. The given bounds clearly demonstrate the advantage of large sparsity parameter s and give a complete picture for sparse JL tradeoffs. I appreciate the amount of technical effort the authors put to analyze sparse JL transforms. The paper seems technically sound and is clearly written. I recommend acceptance.

Reviewer 3



The interesting part of the paper is a new analysis on the same quantity of the embedding error that has been repeatedly studied in the previous works. The previous approaches fail to generalize to either s > 1 or vectors with a general ell_infty/ell_2 bound. Previously the quantity of embedding error is usually treated as a quadratic form in Rademacher variables and the Hanson-Wright inequality can be applied to obtain moment bounds. However, this does not give tight bounds for v(m,eps,delta,s). Instead, this paper uses a more specific inequality on Rademacher quadratic form from Latala, and a bound on the weighted sum of symmetric random variables (originally proved by Latala, while the authors provided an alternate proof). The rest seems laborious case analysis and I wasn't able to verify everything. The author clearly intended to connect this sparse-JL problem to machine learning and thus branded “feature vectors” in the main body of the paper, but the connection is not really not explained. I would recommend the author to drop this superficial phrase and make it a cleaner math paper. Minor comments: - Do not need to write the subscript e for natural log. Use ln instead or just say that log stands for natural logarithm, as this does not affect the asymptotic orders. - Notation inconsistency: the distribution is mathcal{A} in the main body but \mathscr{A} in the supplementary - Last line of the statement of Lemma 3.2, change “...>=...” to “>= … >=” - First line below Section 4: missing a right bracket ) in ||Z_r(...)||_q. There are other places of missing closing brackets, please check the paper thoroughly Lemma 6.1: Change “x_i <= \nu” to “|x_i| <= \nu” - Page 12 of supplementary material: in the bottom equations, what is the supremum taking over? The condition is not clear. Is it T >= max{...,...,...}? This error occurs through the proof. - Lemma 8.1: Isn’t the first sentence redundant?

[Author Response · NeurIPS 2019]

Thank you very much for your reviews.

**Reviewer 1:**

1. Regarding the parameters in Section 4.2, I ran my experiments with 2 more sets of parameters, where the target dimensions are much lower than in Figure 6. The trends match trends in the submission as expected.

Figure 1: Phase transitions of $\hat{v}(m, 0.2, 0.01, s)$          Figure 2: Phase transitions of $\hat{v}(m, 0.05, 0.05, s)$

2. Regarding experimental design in Section 4.2, recall my goal was to compute an estimate $\hat{v}(m, \epsilon, \delta, s)$ of $v(m, \epsilon, \delta, s)$. As mentioned in footnote 20, the design is based on Section 3.1 of [13] (for $s = 1$).

   Here is a more detailed explanation of how $\hat{v}(m, \epsilon, \delta, s)$ is computed: I test using a set $W$ of values $w$ spaced between $0.03$ and $1$ (see footnote 19) . For *each* $w \in W$, I compute $\hat{\delta}(s, m, \epsilon, w)$, an estimate of failure probability on the specific binary vector $x^w$ where the first $1/w^2$ entries are nonzero. Then, I let $\hat{v}(m, \epsilon, \delta, s)$ be the max. value in $W$ such that $\hat{\delta}(s, m, \epsilon, w) \leq \delta$ for all $w \in W$ where $w \leq \hat{v}(m, \epsilon, \delta, s)$.

   *How do I compute $\hat{\delta}(s, m, \epsilon, w)$?* As mentioned in the submission, I estimate $\mathbb{P}_{A \in \mathcal{A}_{s,m,n}}[\|Ax^w\|_2 \notin (1 \pm \epsilon) \|x^w\|_2]$ by computing the projected norm for $T = 100,000$ samples of a block sparse JL matrix.

   *Why does it suffice to only consider sparse vectors $x^w$, rather than all vectors in $S_v$?* As mentioned in footnote 21, I show in the proof of Theorem 1.5 that asymptotically, if a "violating" vector (i.e. $x$ s.t. $\mathbb{P}_{A \in \mathcal{A}_{s,m,n}}[\|Ax\|_2 \notin (1 \pm \epsilon) \|x\|_2] > \delta$) exists in $S_v$, then there's a "violating" vector $x^w$ for some $w \leq \Theta(v)$. Thus, the estimate $\hat{v}(m, \epsilon, \delta, s)$ will approach $v(m, \epsilon, \delta, s)$ up to constants as $T \to \infty$ and as precision in $W$ goes to $\infty$ (if $\epsilon, \delta$ are sufficiently small for the "violating" vector asymptotics to kick in).

3. Regarding the Section 4.1 experiment, the failure probability actually increases to a local maximum somewhere in $12 \leq s \leq 16$, and then decreases when $s \geq 16$, reaching lower than the value at $s = 8$ by the time $s = 20$. When $\epsilon = 0.07$ and $m = 500$, there is similarly a local maximum (somewhere in $24 \leq s \leq 32$) followed by a decrease. The phenomenon of non-monotonicity in $s$ can also be observed on synthetic data in Figure 6 in the submission: for example, when $\delta = 0.05$, $\epsilon = 0.02$, $m = 12000$, we see that $\hat{v}(m, \epsilon, \delta, 4) < \hat{v}(m, \epsilon, \delta, 3)$. I'd like to emphasize that my asymptotic theoretical results characterize the macroscopic behavior of $v(m, \epsilon, \delta, s)$, and do not preclude the existence of constant factor fluctuations for small changes in parameters.

4. Regarding [29], I will certainly add this reference – thanks for pointing this out. I agree that this reduction gives lower bounds for a distribution similar to the uniform sparse JL distribution. However, as you mentioned, the lower bounds differ from my work in the following ways: (a) they do not match the bounds in Theorem 1.5, since they would not recover the branch 3, and (b) the distribution resulting from this reduction gives $s$ nonzero entries that are independently selected (potentially resulting in a multiset), which is different than the uniform sparse JL distribution. Regarding (b), in fact, Theorem 5.1 in the JACM version of [18] shows that this "multiple hashing" distribution requires an extra (roughly) $\log(1/\delta)$ factor on the sparsity to satisfy (1).

5. Regarding obtaining a fine-grained understanding of each vector in $\mathbb{R}^n$, I agree this would be an interesting result, though it would not immediately follow from the techniques that I present in this submission.

**Reviewer 3:**

1. I appreciate that you read through my supplementary material, and I will certainly address the typos you noted. (Regarding the specific typo on p. 12, it should say $T \geq \max(\frac{s\epsilon}{mv^2}, 3), T \leq \log(Tmv^2/s)$.)

2. Regarding your comment about feature vectors, I agree that considering vectors with restricted $\ell_\infty$-to-$\ell_2$ norm ratio is also interesting in its own right from a theoretical perspective. This perspective also does nicely motivate my lower bound on dimension-sparsity tradeoffs (footnote 3, Corollary 1.3 in the supp. material). Nonetheless, my experiments in Section 4.1 link my analysis to feature vectors by considering the performance of sparse JL on feature vectors. I specifically evaluate the performance of sparse JL on bag-of-words feature vectors in two real-world datasets: News20 and Enron emails. Note that [29] also evaluates on bag-of-words datasets (in collaborative spam filtering, and the work initiates the study of vectors with restricted $\ell_\infty$-to-$\ell_2$ norm ratio in this context). Subsequent work [13, 10] also experimentally considers the performance of feature hashing on bag-of-words feature vectors from News20 and a collection of NeurIPS papers.

[Meta-Review · NeurIPS 2019]

Reviewers are all in agreement that this is a very strong, obvious accept. [This meta-review was reviewed and revised by the Program Chairs]